# Teaching Algorithmic Reasoning via In-context Learning

## Abstract

Large language models (LLMs) have shown increasing in-context learning capabilities through scaling up model and data size. Despite this progress, LLMs are still unable to solve algorithmic reasoning problems. While providing a rationale with the final answer has led to further improvements in multi-step reasoning problems, Anil et al. (2022) showed that even simple algorithmic reasoning tasks such as parity are far from solved. In this work, we identify and study four key stages for successfully teaching algorithmic reasoning to LLMs: (1) formulating algorithms as skills, (2) teaching multiple skills simultaneously (skill accumulation), (3) teaching how to combine skills (skill composition) and (4) teaching how to use skills as tools. We show that it is possible to teach algorithmic reasoning to LLMs via in-context learning, which we refer to as *algorithmic prompting*. We evaluate our approach on a variety of arithmetic and quantitative reasoning tasks, and demonstrate significant boosts in performance over existing prompting techniques. In particular, for long parity, addition, multiplication and subtraction, we achieve an error reduction of approximately 10x, 9x, 5x and 2x respectively compared to the best available baselines.

## 1 Introduction

Large language models (LLMs) have shown impressive progress in recent years, driven by the scaling up of models and training data sizes (Kaplan et al., 2020; Wei et al., 2022a; Hoffmann et al., 2022) that has led to improved performance and sample efficiency (Brown et al., 2020; Chen et al., 2021; Chowdhery et al., 2022). One area with significant room for improvement is the ability of LLMs to perform complex reasoning tasks. In this realm, mathematical reasoning (Saxton et al., 2019) provides a unique challenge as a domain. It requires the ability to parse, to logically deconstruct a problem into sub-problems and recombine them, and to apply knowledge of rules, transformations, processes, and axioms.

The idea of providing a rationale with the final answer was first proposed by Ling et al. (2017) and recently revived for LLMs in the form of scratchpad (Nye et al., 2021) and chain-of-thought (Wei et al., 2022b). It has led to improvements in performance on multi-step reasoning problems (Wang et al., 2019) such as arithmetic, commonsense, and symbolic reasoning tasks (Nye et al., 2021; Wei et al., 2022b; Lewkowycz et al., 2022a; Wang et al., 2022a;b; Anil et al., 2022; Zhou et al., 2022). However, despite significant progress, these models still struggle with out-of-distribution (OOD) generalization on reasoning tasks (Nogueira et al., 2021; Kim et al., 2021; Anil et al., 2022).

To successfully generalize out-of-distribution on many of these reasoning tasks, the model needs to learn the underlying algorithm for solving a task. We refer to this behavior as *algorithmic reasoning* (Kaiser and Sutskever, 2015; Veličković and Blundell, 2021). While following an algorithm can be seen as a form of instruction following, algorithms are generally more complex with a larger number of steps, though each step of the algorithm may be simpler and more concise than typical instructions. The benefit of being able to learn algorithms is that since they are input independent by nature, they are immune to OOD performance degradation when executed properly. Moreover, algorithms can be specified without ambiguity and hence provide a good test bed to probe model capabilities.

One surprising capability of LLMs is in-context learning (Brown et al., 2020), which refers to the ability to learn a task from a few examples being presented within a *prompt*. In-context learning does

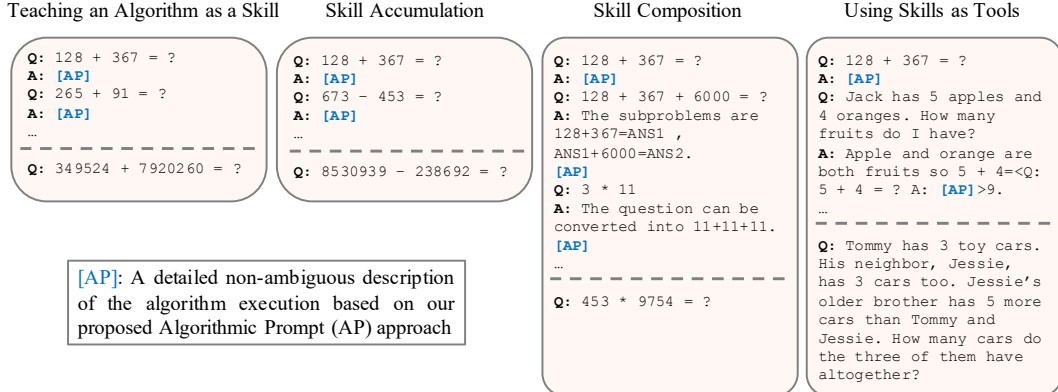

Figure 1: The four learning stages investigated in this work (from left to right): (i) Teaching an algorithm as a skill (Section 3) (ii) Skill Accumulation, i.e., teaching multiple skills simultaneously (Section 4) (iii) Skill Composition, i.e. the ability to learn a complex skill through building upon simpler ones (Section 5) (iv) Using Skills as Tools to solve problems (Section 6). We teach these algorithms in-context using our proposed *algorithmic prompting* approach, which does not involve any further training of the underlying model.

not require any weight updates, and provides a powerful platform for specialized skill acquisition without losing the generality of the underlying model. Moreover, various prompting strategies have shown significant potential in solving certain types of reasoning problems (Jung et al., 2022; Zhou et al., 2022; Wei et al., 2022b; Kojima et al., 2022). Nonetheless, Anil et al. (2022) considered two algorithmic reasoning tasks and showed that while rationale-based prompting allow LLMs to generalize to longer problem instances, they are still far from solving simple algorithmic tasks such as parity.

In this work, we investigate how to teach algorithms and compositions of algorithms to LLMs via in-context learning. This setup is reminiscent of how similar skills are taught to children in school. We identify and explore four key stages for teaching algorithms as skills to LLMs (Figure 1). We begin by studying the shortcomings of existing approaches and proposing ways to alleviate them. We focus on arithmetic algorithms such as addition, subtraction and multiplication as they have been widely benchmarked (Saxton et al., 2019; Hendrycks et al., 2021) and famously fail at out-of-distribution generalization even for the best performing models on the MATH benchmark (Lewkowycz et al., 2022b). While one can avoid learning these algorithms by using external tools such as a calculator (Cobbe et al., 2021), this approach cannot scale to higher levels of abstraction where a model needs to use "soft algorithms" and certain steps must be flexibly applied in different situations.

**Contributions:** Our main contributions are as follows:

- We introduce *Algorithmic Prompting*, which involves providing a detailed description of the algorithm execution on running examples, and using explicit explanation and natural language instruction to remove ambiguity. For a comparison of algorithmic prompting to existing prompting techniques, see Section 2 and Table 1.

- We demonstrate that algorithmic prompting significantly outperforms existing prompting techniques on several algorithmic tasks. In particular, for long parity, addition, multiplication and subtraction, we achieve an error reduction of approximately 10x, 9x, 5x and 2x respectively compared to the best available baselines (Section 3 and Table 2).

- Our ablation studies reveal the impact of non-ambiguous explanations, and show that unlike other prompting approaches, errors in the algorithmic examples affect performance significantly (Section 3.1).

- We study the model's ability to simultaneously learn multiple algorithms via a single prompt, as well as its ability to compose the learned algorithms in order to solve more complex tasks (Sections 4 and 5).

- We explore various approaches to leverage a learned algorithm as a tool to solve math word problems. We show that while it is possible to improve the performance in settings that require complex calculations, the model's general reasoning capability reduces due to the phenomenon of *interference* (Section 6).

## 2 ALGORITHMIC PROMPTING

Nye et al. (2021) proposed the idea of getting the model to show its work, i.e, breaking the problem down and asking the model to output the intermediate steps used in solving the task. The authors show that by finetuning LLMs on such data – which they refer to as *scratchpad* – they can greatly improve performance on multi-step computation problems. This was taken further by Wei et al. (2022b) to the in-context learning setting, where they showed that providing rationales in the prompts significantly increases the model's ability to solve multi-step reasoning problems. They refer to this approach as *chain-of-thought*. The main intuition behind the scratchpad approach is that by having intermediate computations in the output, the model can refer to them directly instead of relying on its internal representation space for those calculations. For chain-of-thought, one hypothesis is that the rationales loosely provide the model with a "thinking pattern" that it can reference when tackling a problem. By encouraging the model to output an explanation along with the answer, we steer it towards solving problems by breaking them into steps that logically follow from each other. Inspired by these perspectives, we hypothesize that if we increase the specificity and applicability of these thinking patterns, we can also increase the amount by which the model adheres to these patterns in its problem solving. As we will illustrate, this approach leverages both the scratchpad ideas of showing intermediate computations and the chain-of-thought ideas of providing an explanation for each step.

As a motivating example, consider the standard addition algorithm. This method right-aligns the two numbers being added and calculates the sum of pairs of single digits from each number, going from right to left. For every pair of digits, there is a possible carry that needs to be added to the next digit sum. If we use a scratchpad-style illustration, then for a question like $182 + 376$, the model would see that the digit-sum $2+6$ generates a carry of $0$, while $8+7$ generates a carry of $1$. However, the rule for how carry should be computed is highly ambiguous from just this example. Ideally, we expect the model to conclude that for $x + y = z$, the carry should be computed as $c = (z - (z \mod 10))/10$. But from only this example, the model could conclude that the carry is $1$ whenever we add two even digits together and $0$ otherwise, or that the first digit-pair generates a carry of $1$, the second digit-pair generates a carry of $0$, and so on. In order for the model to extrapolate the correct pattern, it must be biased in such a way that the general and correct rule is the default interpretation. Such alignment, however, can not be reliably expected from current models.

We hypothesize that existing prompting methods fail to sufficiently constrain the model's interpretation of the prompting information, and result in unexpected and undesirable model behaviors on tasks that require precise algorithmic reasoning. We push the limits of rationale-based prompting by drastically increasing the amount of detail included in the rationales, while specifying the steps of an algorithm within this information. We refer to this strategy as *Algorithmic Prompting*, and contrast this approach with other types in Table 1. We show that it can achieve significant systematic generalization on several algorithmic reasoning tasks, and ground our exploration in the four capabilities identified in Figure 1.

Table 1: Comparison of different prompting strategies studied in this work. The number of ★ indicates the level to which each strategy exhibits the given characteristic. In this work, we refer to the basic approach of presenting only input-target pairs with no additional explanation as *few-shot*, and we refer to prompts that provide explicit instructions but no running examples of a task as *instruction-only*. We see that algorithmic prompt includes both qualities of natural language explanation and explicit intermediate computations.

| Prompt strategy | Input-target pairs | Natural language rationale | Intermediate computations | Rationale diversity |
|---|---|---|---|---|
| Few-shot | ★ ★ ★ | - | - | - |
| Chain-of-thought | ★ ★ ★ | ★ ★ ★ | ★ | ★ ★ ★ |
| Scratchpad | ★ ★ ★ | - | ★ ★ | - |
| Instruction-only | - | ★ ★ ★ | - | ★ ★ ★ |
| Algorithmic | ★ ★ ★ | ★ ★ | ★ ★ ★ | ★ |

### 2.1 EXPERIMENTAL SETUP

**Baselines:** We compare the proposed algorithmic prompt to few-shot and chain-of-thought baselines in our experiments. The *few-shot* baseline refers to the simple approach of presenting examples of question and answer pairs with no additional explanation. The *chain-of-thought* baseline provides

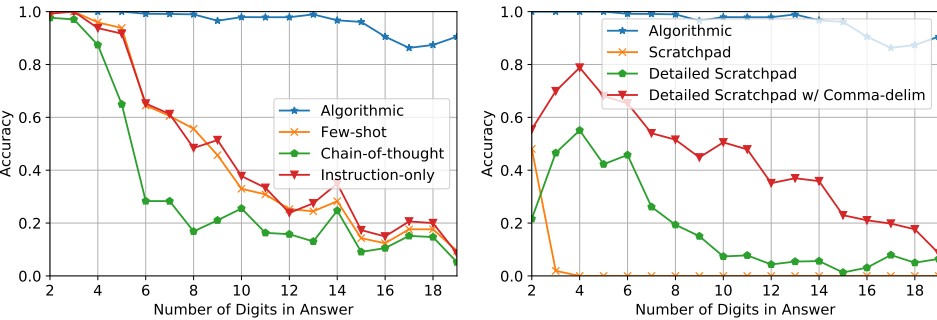

(a) Various prompting strategies on addition  (b) Variants of scratchpad prompting on addition

Figure 2: Accuracy on addition questions of increasing length for different prompting methods. Addition questions are of the form $a + b = c$, where $a$, $b$, and $c$ are positive integers. The number of digits in answer plotted in the x-axis refers to the length of $c$. Accuracy is measured over 2000 total examples sampled uniformly over the length of $c$. The max length for examples in the prompt is 5. **Left:** We see that algorithmic prompting shows near-perfect length generalization even on extremely long addition questions, and significantly outperforms its simple few-shot and chain-of-thought counterpart. **Right:** using scratchpad-style output as a prompt leads to abysmal performance, but adding a few extra details to the scratchpad format leads to non-trivial generalization.

a rationale along with the final answer in the few-shot examples. In order to generate the rationale for various tasks, we follow the method introduced in Kojima et al. (2022) and use the phrase "let's think step by step" to get a model-generated rationale for the few-shot examples.

**Evaluation metric:** We measure both in-distribution and OOD performance in all experiments. For the in-context learning setting considered in this work, the data distribution is determined by the answer lengths of the prompting examples. Thus, questions with answer lengths that fall within those seen in the prompt are considered in-distribution, and those with longer lengths are considered out-of-distribution. The choice of length is natural given that it is a measure of complexity in the tasks we consider, and length generalization has a rich history as a measure of systematic generalization (Csordás et al., 2021; Anil et al., 2022). Thus, length generalization provides a good indication for whether the model has learned the underlying algorithm.

**Experimental setting:** We use the Codex model `code-davinci-002` from OpenAI (Chen et al., 2021) for all experiments. This model has a maximum context length of 8000 tokens. Task examples are sampled uniformly at each length. All results are sampled once using a temperature of 0 and default settings for other hyperparameters. See Section A.2 for task details.

## 3 TEACHING ALGORITHMS AS SKILLS

### 3.1 TWO-NUMBER ADDITION

We begin our analysis by studying the two-number addition task and explore the effectiveness of various prompting strategies with differing levels of ambiguity. We present an algorithmic prompt for addition and compare its performance against the few-shot, chain-of-thought, instruction-only, and scratchpad methods. An illustration of these prompting strategies is shown in Figure 8, and the prompts can be found in Section B.1. For all addition experiments, we use 3 prompt examples and restrict these examples to having answers of up to 5 digits in length. We then evaluate on questions up to 19 digits in length. The length of 19 is chosen because this is the level after which the model begins to run out of context. A similar choice is used for all algorithms considered in Section 3.

Figure 2(a) shows the performance of algorithmic prompting against existing methods on addition problems. These results demonstrate that algorithmic prompt achieves near perfect performance and OOD generalization on addition, while few-shot, chain-of-thought, and instruction-only have decreasing performance as the length of the answer increases. These results illustrate the benefit of incorporating algorithmic steps, unambiguous explanations, and demonstrations on running examples in our prompt.

**Impact of unambiguous explanations:** Figure 2(b) compares the performance of using scratchpad and detailed scratchpad as prompts. With detailed scratchpad, we add more intermediate steps to illustrate how the values of the answer (A) and the carry (C) is derived (see Figure 8). We further include an additional version that converts the numbers from space-delimited to comma-

delimited, as we observed that comma is a more effective deliminator for Codex. We find that the scratchpad template performs extremely poorly as a prompt[1], but including additional details leads to a significant boost in performance. We conjecture that the abysmal performance of scratchpad as a few-shot prompt is due to the structure of the solution format being sufficiently regimented to move the model away from its memorized solutions, but not clear enough for the model to extract the true underlying rules and adapt them to new examples.

We also compare the algorithmic prompt to two less-detailed variants. One version (*nonexplicit calculation*) omits the explicit equation showing how the carry value is derived. This shares the same intuition as the original motivating example. The second version (*uncommon operation*) requires the model to index the correct digit for a given step. The indexing of a digit at a variable position is a more uncommon operation than the indexing of the digit at the same position each time. In our final addition prompt, we introduce a mechanism that allows the model to avoid the indexing operation by copying the unprocessed digits over to each step and always taking the last digit. Figure 3(a) illustrates the relative gains that come from the disambiguation of these two aspects of the algorithm. Prompts used for the ambiguity ablation studies can be found in Section B.2.

**Is the model actually learning the algorithm through in-context learning?** Min et al. (2022) have shown that it is not necessary to provide the correct question-answer pairings in the few-shot prompt, suggesting that the model does not rely on the demonstrations themselves to figure out the right way to solve the given task. However, in order to claim that we are teaching algorithms in-context, we would like to understand whether the model is actually following the algorithm as it is prescribed in the prompt. To do so, we validate that 1) mistakes in the intermediate output steps lead to mistakes in the final answer, and 2) errors in the prompt significantly impact performance.

We first look at the errors that the model makes. We find that for every question where the final answer was correct, *all* intermediate steps were also correct. Next, we analyze the performance of the model when we introduce errors into the algorithmic steps in the prompt. We introduce errors into the second digit of the calculation step ($digit_1 + digit_2 + \text{carry} = \text{answer}$), and keep everything else the same. We consider two types of errors: *irregular errors* where only a subset of the steps contain an error, and *systematic errors* where all of the steps presented in the prompt contain an error. With irregular errors (prompt shown in Section B.2.3), the model still has a chance of extrapolating the correct rule based on the unchanged steps. With systematic errors (prompt shown in Section B.2.4), the model should *not* derive the correct rule if it was truly learning from context, rather than simply mapping to the output format and overriding the in-context steps with what it knows from pretraining. Figure 3(b) shows that there is a small degradation in performance with irregular errors, while the accuracy drops to near $0\%$ with systematic errors, thus confirming the expected behavior of a model that is actually learning in-context. This is in contrast to the findings in which providing shuffled targets (Min et al., 2022) or wrong patterns in chain-of-thought (Madaan and Yazdanbakhsh, 2022) do not materially impact model's performance. Thus, algorithmic prompting differs from other approaches and constrains the model's behavior towards what is actually being taught in-context.

## 3.2 TEACHING OTHER ALGORITHMS USING ALGORITHMIC PROMPTING

To validate that the performance of algorithmic prompting is not specific to two-number addition, we evaluate model performance on subtraction, multiplication, and parity tasks. Similar to addition, the maximum length evaluated is based on the length that can fit into context for algorithmic prompts.

**Subtraction:** We follow a similar strategy as addition. We discuss the peculiarities of the subtraction algorithm in more detail in Section 4, where we combine both addition and subtraction problems. The performance at length $14$ is summarized in Table 2. We see that algorithmic prompting significantly outperforms the few-shot baseline.

**Multiplication:** The multiplication algorithm requires $O(n^2)$ steps if we use a strategy similar to the addition algorithm which takes $O(n)$ steps. Inspired by this complication, we explore whether the model's existing zero-shot or few-shot capabilities can be leveraged in conjunction with algorithmic prompting to reduce the complexity of the required instructions. Therefore, instead of using

---

[1]The original paper by Nye et al. (2021) performs finetuning using the scratchpad format, whereas we directly use it as a prompt.

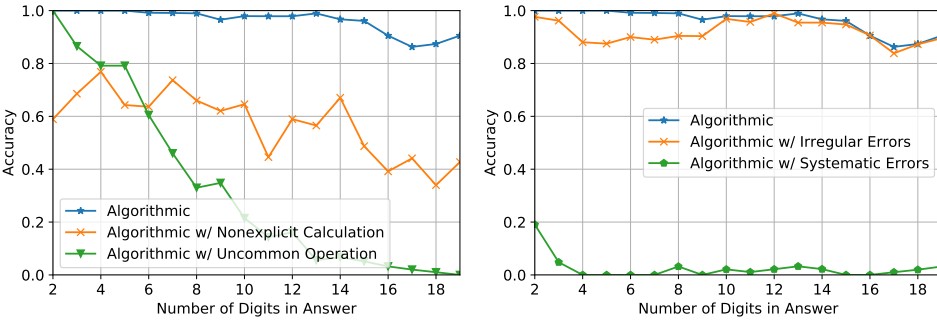

(a) Algorithmic prompts with varying ambiguity          (b) Algorithmic prompts with errors

Figure 3: Accuracy on addition questions of increasing length for variants on the algorithmic prompt. **Left:** Two examples of rule ambiguity that we address in the final addition prompt are non-explicit carry calculation (*Nonexplicit Calculation*) and digit indexing (*Uncommon Operation*). We observe a significant difference in performance before and after reducing the ambiguity of these operations. **Right:** Errors are introduced to the algorithmic prompt in the digit value of the second number in the equation. *Irregular errors* are introduced to a minority subset of steps in the algorithmic examples, while *systematic errors* are introduced to all steps of the examples. We see that irregular errors have a minor impact on the performance, and systematic errors completely destroy the model's ability to solve this task. This suggests that the model is following the algorithm as it is specified in-context, rather than loosely mimicking the format of the algorithm.

single-digit multiplication in each step, we perform direct calculations for 1-digit $\times$ $n$-digit numbers. Instead of doing $n^2$ single-digit calculations for two $n$-digit numbers, we now only need to perform $n$ steps of $1 \times n$-digit multiplication. More details can be found in Section A.4. Performance at length 7 is shown in Table 2, and performance across different lengths is shown in Figure 4. We see that the multiplication algorithmic prompt performs well compared to its few-shot and chain-of-thought counterparts, thus illustrating the potential of utilizing a model's inherent abilities within the scaffolding of more structured algorithmic instructions.

**Parity:** We consider the problem of calculating parity of a given binary list. This task has been studied extensively in Anil et al. (2022), and despite the intrinsic simplicity of the algorithm, it is far from being solved. Performance at length 20 is shown in Table 2. We see that algorithmic prompt significantly outperforms random chance on this task, which is even greater than the few-shot performance reported in Anil et al. (2022). More details can be found in Figure 12 and Section A.4.

Table 2: Performance on addition, subtraction, multiplication, and parity tasks. For addition we use the few-shot baseline and evaluate at length 19. For subtraction we use the few-shot baseline and evaluate at length 14. For multiplication we use the chain-of-thought baseline and evaluate at length 7. These lengths are chosen based on the maximum task length that could fit into context for the algorithmic prompt. For parity we evaluate at length 20 which is the longest instance reported in Anil et al. (2022).

| Method | Addition | Subtraction | Multiplication | Parity |
|---|---|---|---|---|
| Algorithmic prompt | 90.5% | 65.6% | 79.7% | 95.0% |
| Best available baseline | 9.5% | 16.7% | 5.5% | 50.0% |

## 4 SKILL ACCUMULATION

So far we have demonstrated the ability to teach single-algorithms through in-context learning. In this section, we study the model's ability to simultaneously learn multiple algorithms and choose the applicable one when solving problems, which we refer to as skill accumulation. To do so, we use the addition-subtraction task. We expand on the addition problem to allow for both positive and negative numbers. Thus, the problems now have four possibilities: $a + b$, $-a + b$, $-a - b$, $a - b$. We refer to questions of the form $a + b$ as *addition-only* questions, and the rest as *subtraction-only* questions. For subtraction questions, the ordering of the two numbers matter. To see this, consider the examples $43 - 250 = -207$ and $543 - 250 = 293$. When we process the digits from right to left, the answer depends on whether the first number is greater than or less than the second number in absolute value, not just on the values of the two digits. Thus, subtraction requires a different – albeit similar – algorithm to addition. For a sense of the relative complexity of the two settings, note that the subtraction algorithm we use runs in $2n$ steps, while the addition algorithm runs in $n$ steps.

To succeed at this task, the model needs to demonstrate the ability to follow different processing paths when the question is addition or subtraction. Figure 5 shows the performance of the combined

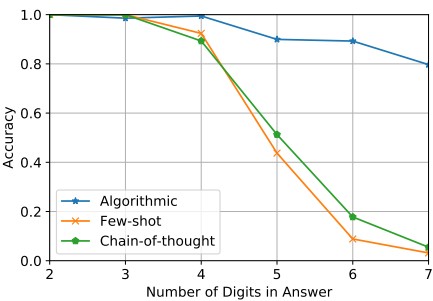 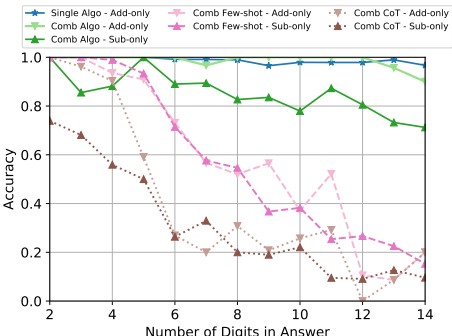

Figure 4: Performance of algorithmic prompt on multiplication questions, where at least one of the two numbers in the question is less than 1000. We use 2 shots of up to 6-digits in answer length in all the prompts. Algorithmic prompt shows superior length generalization compared to the baselines.

Figure 5: Accuracy on addition and subtraction questions using a combined prompt. We use 6 shots of up to 5-digits in length in the prompt (2 shots of addition and 4 shots of subtraction). "Comb Algo" refers to the combined algorithmic prompt with both addition and subtraction examples, while "single algo" refers to the algorithmic prompt for addition in Section 3.1.

addition-subtraction prompt, with the accuracy broken down by question type. We see that the model is able to effectively execute the correct algorithm based on the individual questions. The model exhibits lower accuracy on subtraction questions compared to addition-only questions, reflecting the increased complexity of the subtraction algorithm. Comparing the performance on addition-only questions to the addition prompt from Section 3.1, we see that there is minimal change in performance despite having an extra other algorithm present in the prompt. More analysis regarding the effects of combining these two algorithms can be found in Section A.5.

## 5    SKILL COMPOSITION

In this section, we explore the model's ability to learn multiple algorithms that build on top of each other. This is a desirable property because it enables the model to learn a more complex algorithm without having to relearn simpler sub-components of that algorithm and enables modularization of complex algorithms. To establish a framework for skill composition, we explore two extensions to the addition algorithm: 1) adding multiple numbers together, and 2) solving multiplication by turning it into an addition problem (e.g. by converting $3 * 7$ into $7 + 7 + 7$). The ability to add multiple numbers builds on top of the ability to add two numbers together. Solving multiplication as addition further builds on the addition of multiple numbers. An illustration can be found in Figure 15. The evaluation dataset contains 1000 examples sampled uniformly by length of answer.

The performance on composite tasks are shown in Figure 6. We teach these algorithms in-context by creating a composite prompt that includes 2 examples from the 2-number addition prompt, 1 example of addition of 3 numbers, and 1 example of converting multiplication into addition. This forms a simple composition strategy (*Algo - (Simple Comp)*). This prompt can be found in Section B.7. We also consider two ablations of the composite algorithmic prompt. The algorithm for $n$-number addition involves wrapping 2-number additions within a larger loop of $n - 1$ addition problems. Thus, we could provide even more information by converting the 2-number addition prompt examples into the same loop format as the 3-number addition example. This version (*Algo - (Augmented Comp)*) provides an upper estimate on multi-number addition and multiplication-as-addition. The second ablation (*Algo - (No Comp)*) only presents the example that illustrates the extended skill. This has no composition and provides a lower estimate on the performance of the two extended skills, and illustrates the improvement that comes from having first learned the component algorithms. See Figure 16 for an illustration of the different composition strategies.

In-context skill composition is limited by the context length of current models. Unlike the previous experimental results, these composition tasks include a number of questions that were incomplete for the algorithmic prompt. To separate out the issue of context length from the ability of the model to follow an algorithm, in Figure 6 we report performance on only the questions for which the algorithmic prompt could fit into context. This subset is also used for all baselines. Figure 6 shows that the algorithmic prompt significantly outperforms few-shot and chain-of-thought baselines. Moreover, we observe that there is minimal difference between the simple composition and augmented composition strategies, and that the "no composition" approach performs much worse than its composed counterparts. We also explore two approaches to move beyond context length limitation: 1) a

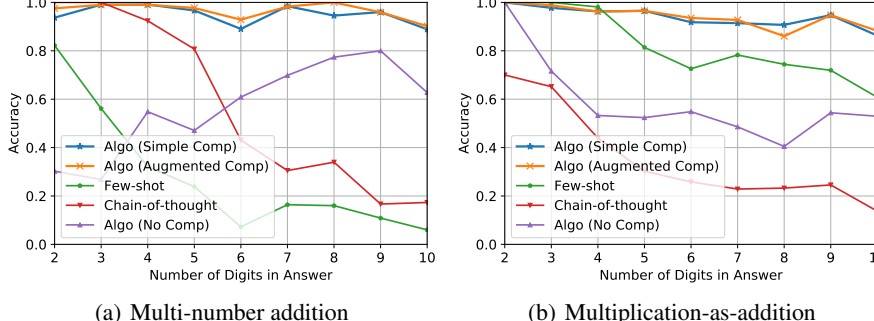

(a) Multi-number addition  (b) Multiplication-as-addition

Figure 6: Performance on compositions of skills. "Algo" indicates algorithmic prompting. "Simple Comp" refers to a simple composition strategy where previously taught algorithms are transferred as is. "Augmented Comp" adjusts the previously taught algorithm to match the format of the new task. This simulates a version where the full prompt specializes to the new task. "No Comp" uses only the part of the "Simple Comp" prompt that describes the new task. This simulates the comparison to learning a new skill from scratch without first learning its stepping stones. We observe that the composed algorithmic templates demonstrate better generalization than the baseline methods. Note that for multiplication, we evaluate the algorithmic methods on a harder task than the few-shot baselines, since we force the model to convert the question into the addition of a number $n$ times, while for other baselines we simply perform $1 \times n$-digit multiplication directly.

second-pass approach where only the last completed step instead of the full output is used during a second inference pass, and 2) a dialogue-like strategy where separate contexts are used for individual subroutines of the algorithm. We describe these approaches and their results in detail in Section A.6. Overall, we find that we can achieve performance comparable to Figure 6 on the full dataset.

## 6  USING SKILLS AS TOOLS

In this section, we study the behavior of the model when using a given algorithm as a step in solving a larger mathematical reasoning problem. Such problems (e.g GSM8k benchmark (Cobbe et al., 2021)) usually consist of two components: 1) the informal mathematical reasoning component which requires the model to come up with the correct solution steps to arrive at the answer based on the information provided in the question and 2) the calculation of arithmetic operations used in the solution steps. In this paper, we study how the model can leverage a learned algorithm to improve the quality of the second component, i.e., arithmetic operations inside a broader reasoning process. Although an external calculator can be used in this case, this will not be possible in general for more abstract skills such as simplifying mathematical equations.

**Dataset**:  We consider the following two math word problem datasets: **GSM8k** and **GSM8k-Hard**. GSM8k (Cobbe et al., 2021) consists of high-quality mathematical reasoning problems presented as natural language questions. In order to study the ability to use the addition algorithm while solving GSM8k questions, we simplify the task by filtering for a subset of GSM8k whose solutions consist of only addition steps. The filtering procedure results in $108$ pure-addition GSM8k questions. To further illustrate the potential of leveraging skills as a form of tool use, we create a hard dataset called GSM8k-Hard, which consists of $50$ examples from the pure-addition subset of the GSM8k with larger numerical values. For more detail on the datasets, see Section A.7.

We first evaluate whether the chain-of-thought prompt can be augmented with the algorithmic prompt for the addition operation. To do so, we use a single prompt to illustrate both the informal mathematical reasoning skill and the addition skill. Specifically, we embed the addition algorithm within the chain-of-thought solutions whenever the solution calls for the summing of numbers. We use an `<ALGO>` flag within the prompt to indicate that algorithmic reasoning should be used, and the `<NONALGO>` flag to indicate direct, non-algorithmic calculations. Results are shown in Figure 7(a). A detailed description of this prompting strategy and the corresponding results can be found in Section A.7.1. We find that including algorithmic output in the examples significantly disrupts the model's informal mathematical reasoning abilities in the `<ALGO>` experiment, but leaves the `<NONALGO>` performance relatively unchanged. This demonstrates the existence of interference between the two skills. This is in contrast to the results from skill accumulation in Section 4. We conjecture that this occurs when we mix highly different skills within the same context.

Next, motivated by context length limitations and the interference issue that we have identified, we propose a way to alleviate these problems through a dialogue-like interaction between models loaded with different prompts. We do so by teaching the model to output specific tokens for when the addition algorithm should be called. See Figure 7(b) for an example. See Section A.7.2 for a de-

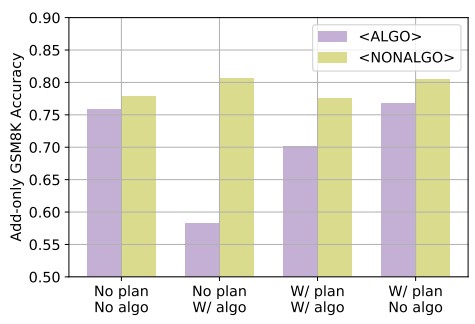

| (a) Performance on GSM8k addition-only subset | (b) Example of tool use for GSM8k-Hard. |
|---|---|

Figure 7: Addition algorithm as tool use in solving GSM8k questions. **Left:** Ablation study with or without algorithmic output in the prompt or output. "W/ algo" indicates that algorithmic output is embedded within prompt examples, and "no algo" indicates that only chain-of-thought rationale is included in the prompt. `<ALGO>` flag indicates that algorithmic reasoning is encouraged in the output, while `<NONALGO>` flag indicates that calculations are done directly by the model. "Plan" indicates a chain-of-thought strategy that summarizes the solution plan before executing individual reasoning steps. We see that having algorithmic output within context leads to significant interference with the model's informal mathematical reasoning abilities. This is alleviated by using a summary before the algorithmic output, but not fully. **Right:** An example question-answer pair from the GSM8k-Hard addition dataset, which includes GSM8k-like questions with large numerical values.

tailed description of this approach. The performance on GSM8k-Hard is shown in Table 3. Logical accuracy refers to the correctness of the solution setup, while addition accuracy refers to the correctness of the calculations steps within the solution. We see that despite removing the algorithm output from the context of the first model, we still observe interference coming from the use of specific tokens in the informal natural language solution steps. Nonetheless, the method that leverages algorithmic tool use still achieves double the accuracy as the baseline chain-of-thought method without algorithmic prompting. Lastly, this result illustrates the ability of dialogue-based tool use to bypass context length limitations, as a single model would not have fit all the output within its context.

Table 3: Performance on GSM8k-Hard addition dataset with or without algorithmic tool use. We see that the overall performance is doubled when we call on a second model loaded with the algorithmic addition prompt to perform addition calculations, demonstrating the potential of leveraging in-context algorithmic skills as a form of tool-use. Moreover, we observe that this performance gain comes directly from more accurate addition accuracy, and the model that performs informal mathematical reasoning still suffers from interference due to the use of specific tokens in the logical reasoning output, as shown in the decreased logical accuracy.

| Method | Overall Accuracy | Logical Accuracy | Addition Accuracy |
|---|---|---|---|
| Chain-of-thought w/ Algo call | 55.8% | 57.7% | 98.4% |
| Chain-of-thought wo/ Algo call | 27.4% | 70.6% | 61.9% |

## 7    CONCLUSION AND FUTURE WORK

Motivated by the potential of in-context learning as a general mechanism for compositional skill acquisition in LLMs, we studied teaching algorithmic reasoning via in context learning. We identified and studied the fundamental building blocks towards this goal and investigated four settings: teaching an algorithm as a skill, skill accumulation, skill composition and using skills as tools. We investigated the shortcomings of existing approaches and proposed algorithmic prompt to alleviate them, showing that it leads to significant performance boost in various algorithmic reasoning tasks. Our work suggests that it may be possible to convert longer context length to better reasoning performance by providing more thorough explanations. This highlights the ability to leverage long contexts (either through increasing context length or other means such as implementing recurrence or an external memory) and generate more informative rationales as promising research directions.

We identified the interference phenomenon for tool use application and investigated different ways to reduce its effect. Our observations about interference suggest that teaching the model the ability to retrieve or selectively attend to specific instructions when solving the particular problem is an important future direction. Moreover, given that there are ongoing efforts in the community to increase the context length of LLMs, it is of interest to design more challenging tasks for each of the four introduced settings and investigate what capabilities can be taught to LLMs when having access to extremely large context length.

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

# A APPENDIX

## A.1 ADDITIONAL RELATED WORK

Mathematical reasoning (Chiang and Chen, 2018; Saxton et al., 2019) has been subject of interest for a long time. Faldu et al. (2021) summarizes the mathematical reasoning benchmarks that are in the form of math-word problems. In addition to this class of benchmarks, formal mathematics in the form of theorem-proofs (Rabe et al., 2020; Li et al., 2020; Polu and Sutskever, 2020; Welleck et al., 2021; Jiang et al., 2022; Wu et al., 2022) has been considered extensively. In this work we focus on algorithmic reasoning for arithmetic tasks and solving GSM8k (Cobbe et al., 2021) problems.

Algorithmic reasoning is typically approached via using structured architectures such as graph neural networks(GNNs) and modifying the architecture align to the algorithms under consideration (Kaiser and Sutskever, 2015; Chiang and Chen, 2018; Xu et al., 2019; Gordon et al., 2019; Yan et al., 2020; Chen et al., 2020; Xhonneux et al., 2021; Veličković and Blundell, 2021) or to the input format (Thawani et al., 2021). However, in this work we focus on teaching algorithmic reasoning to general purpose transformer-based (Vaswani et al., 2017) models.

There has been some recent works investigating in-context learning phenomena. Razeghi et al. (2022) showed that the performance of LLMs on mathematical calculations correlates with term frequency in the training data. Min et al. (2022) investigate which parts of the (input, output) pairs in the prompt play a role in model's performance on 12 NLP tasks. Madaan and Yazdanbakhsh (2022) investigate this for chain-of-thought prompts and conclude that the combination of text and patterns together play a role. Jones and Steinhardt (2022) compares failure modes of LLMs to human biases in the context of few-shot prompts.

There have been attempts of improving chain-of-thought or scratchpad-like prompts for two-number addition. Akyurek and Akyurek (2022) provides an analysis of the scratchpad approach to prompting. They studied various style choices and natural language descriptions and evaluated their impact on performance. Their focus is on what type of formatting works best with scratchpad, and show that having certain natural language descriptions within the prompt benefits performance. In contrast to their approach, we focus much less on the formatting and style of the provided explanations. Instead, our key insight is in elucidating certain operations in an unambiguous manner. Moreover, Zhou (2022) and Chen (2022) proposed other variants of the chain-of-thought approach to addition. We compare the performance of these three methods in Figure 23, and we find that all of them significantly underperforms the algorithmic and few-shot baselines. We provide a detailed comparison of these prior works in Section A.8.

## A.2 Additional information on experimental setup

In Table 4, we provide a summary of experimental settings for all arithmetic and parity experiments in this paper.

Table 4: Evaluation setting for arithmetic and parity tasks. Questions are sampled uniformly based on answer lengths, with an average of 100 samples per length. *For composition tasks of multi-number addition and multiplication-as-addition, the number of shots indicate the number of prompt examples that illustrate the particular composition skill, but more examples of other skills may be included in the prompt. See the corresponding sections for more details.

| Task | Length range in eval | No. of shots in prompt | Max length in prompt |
|---|---|---|---|
| Two–number addition | 2 - 19 | 3 | 5 |
| Subtraction (combined) | 2 - 14 | 6 (2 add, 4 sub) | 5 |
| Multiplication | 2 - 7 | 2 | 6 |
| Parity | 2 - 30 | 2 | 8 |
| Multi-number addition | 2 - 10 | 1* | 4 |
| Multiplication-as-addition | 2 - 10 | 1* | 2 |

## A.3 Additional results on two-number addition

This section includes additional details and results for Section 3. In Figure 8, we provide an illustration of different prompting strategies for two-number addition with differing levels of detail in the explanation.

**The role of natural language within algorithmic prompt:** Since the algorithmic prompt leverages both natural language descriptions and intermediate computations, we disentangle the two components and study the role that natural language plays in the algorithmic prompt. To do so, we consider the following ablations: 1) a symbols-only version of the original algorithmic prompt for addition, where we strip away most of the natural language descriptions, but still retain the use of certain keywords such as Len and Max (Section B.2.5), 2) a symbols-only version where keywords Len and Max are replaced with random words VBZ and UXO (Section B.2.6), and 3) a symbols-only version where keywords are replaced with adversarial words Str and Min, which are associated with known other operations in the pretraining distribution. The results of the ablations are shown in Figure 9. We see that there is a small but clear drop in performance when we move from the original prompt to the symbols-only prompt. We observe a further drop when certain keywords are replaced by uninformative symbols. These results point to the usefulness of leveraging the natural language understanding of LLMs in specifying aspects of the algorithm. Moreover, we see that using misleading symbols leads to a significant drop in performance, which further illustrates the model's reliance on its pretraining when interpreting the algorithmic instructions.

**Error analysis:** We perform an error analysis for the results of using algorithmic prompting for two-number addition. Details of various error categories are found in Table 5. We see that the model can reliably perform single-step operations, such as identifying the max number of digits, calculating two-digit sums (with carry), and copying the previous carry value to the next step. However, the

```
Input:
128+367
Target:
<scratch>
1 2 8 + 3 6 7 , C: 0
1 2 + 3 6 , 5 C: 1
1 + 3 , 9 5 C: 0
, 4 9 5 C: 0
4 9 5
</scratch>4 9 5.
```

Scratchpad

```
Input:
128+367
Target:
<scratch>
[1,2,8] has 3 digits.
[3,6,7] has 3 digits.
[1,2,8] + [3,6,7] , C=0 , 8+7+0=15 , A->5 , C->1
[1,2] + [3,6] , A=[5] , C=1 , 2+6+1=9 , A->9 , C->0
[1] + [3] , A=[9,5] , C=0 , 1+3+0=4 , A->4 , C->0
[] + [] , A=[4,9,5] , C=0 , END
</scratch>
4 9 5
```

Detailed scratchpad w/
comma-delim

```
Problem: 128+367=?
Explanation:

Let's think step by
step.

128+367=128+300+67=428+6
7=495.

The final Answer is 495.
```

Chain-of-thought

```
Problem: 128+367=
...
Length of FN is 3. FN=[1,2,8]. Length of SN is 3.
SN=[3,6,7]. FN[3]=8. SN[3]=7. C[3]=0. Since 8+7+0=15,
15%10=5. Length of A is 1. Thus A=[5]. Since (15-5)/10=1,
C[2]=1.
Length of FN is 2. FN=[1,2]. Length of SN is 2. SN=[3,6].
FN[2]=2. SN[2]=6. C[2]=1. Since 2+6+1=9, 9%10=9. Length of
A is 2. Thus A=[9,5]. Since (9-9)/10=0, C[1]=0.
...
There are no more digits and C[0]=0. Thus the process is
complete. The final Answer is [4,9,5].
```

Algorithmic prompt

Figure 8: Examples of the two-number addition prompt using different techniques.

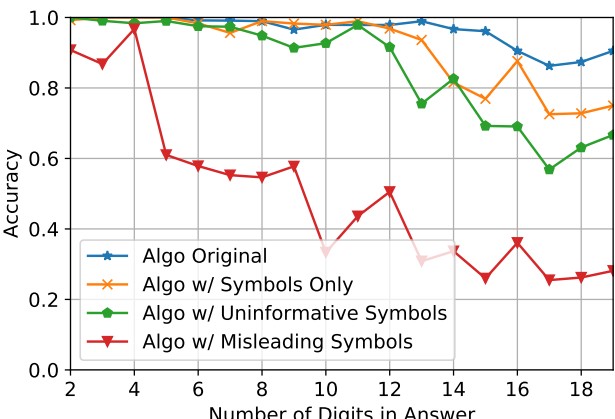

Figure 9: Performance of various symbols-only algorithmic prompts on the two-number addition task. Symbols-only prompt strips natural language from the original prompt, but keeps the use of keywords such as `Len` and `Max`. Uninformative symbols replaces `Len` and `Max` with random words `VBZ` and `UXO`. Misleading symbols replaces `Len` and `Max` with other known words `Str` and `Min`. We see that the symbols-only prompt performs worse than the original algorithmic prompt, and that removing the use of known keywords and replacing them with uninformative symbols results in a further drop in performance. This illustrates the usefulness of natural language descriptions in the prompt. Using misleading symbols leads to a significant drop in performance, further demonstrating the model's reliance on its pretraining when interpreting the algorithmic instructions.

model struggles with multi-step operations such as separating digits by comma and copying all digits within a list from the previous step.

We also see that most of the errors happen in the earlier steps of solving the problem. This is illustrated in Figure 10. The first steps have the most number of unprocessed digits, which may explain why they are the most error prone as the model struggles to copy the lists of digits from step to step.

Table 5: Error analysis of two-number addition algorithmic prompt results. We see that the most error-prone steps are faithfully copying a list from a previous step, followed by counting and separating out digits into list format.

| Error Category | Overall Accuracy | Wrong Questions Only |
| --- | --- | --- |
| Count of first number digits | 99.55% | 88.46% |
| Count of second number digits | 99.04% | 75.64% |
| Identify max number of digits between first and second number | 100.0% | 100.0% |
| Convert first number to list format | 99.6% | 89.74% |
| Convert second number to list format | 99.19% | 79.49% |
| Copy unprocessed digits from first number | 99.55% | 88.46% |
| Copy unprocessed digits from second number | 97.88% | 46.15% |
| Extract last digit from unprocessed first number digits | 99.9% | 97.44% |
| Extract last digit from unprocessed second number digits | 99.85% | 96.15% |
| Copy previous carry value in two-digit calculation step | 100.0% | 100.0% |
| Sum of two digits calculation | 100.0% | 100.0% |
| Calculate new carry value from two-digit calculation step | 99.8% | 94.87% |
| Copy previously accumulated answer digits | 99.14% | 78.21% |
| Insert new value from two-digit calculation result into answer | 99.65% | 91.03% |

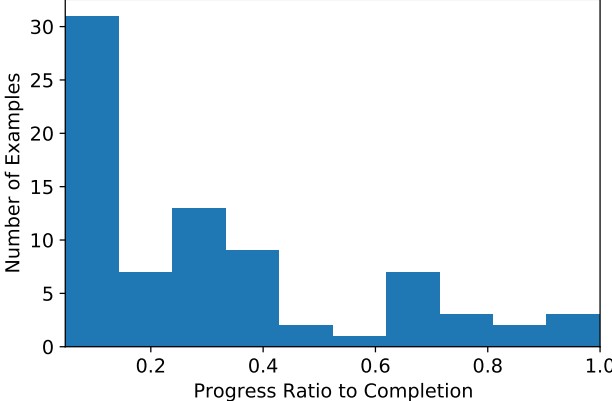

Figure 10: Distribution of where errors first occur in two-number addition questions using algorithmic prompt. Progress ratio is calculated as (first_error_step / total_steps). We see that errors occur in earlier steps, where the model has more remaining digits to process.

A.4    ADDITIONAL RESULTS ON TEACHING OTHER ALGORITHMS

This section includes additional details and figures for Section 3.2.

**Multiplication:**    To choose a reasonable value of $n$ for this experiment, we evaluate the model's zero-shot accuracy in $1 \times n$-digit multiplication (shown in Figure 11). We see that after $n = 3$, the zero-shot performance deteriorates drastically. Thus, we restrict to $n \leq 3$. If a number has more than 3 digits, we break it down into groups of $\leq 3$ digits and add the resulting sub-components appropriately. For simplicity, we consider the problems where at least one of $a$ and $b$ is less than 1000, so that we only need to perform the group splitting on one of the two numbers.

The prompt used for this experiment is displayed in Section B.4.2. We use 2 shots of up to 6-digits in answer length in the prompt. As seen in the prompt, we explain how to break large numbers into groups of 3 or fewer digits in natural language. This natural language description is detailed enough such that the model can correctly extrapolate to creating multiple splits for long numbers, even though it has only seen examples of single splits in the prompt. This illustrates the benefit

of using natural language instructions along with showing the intermediate calculation steps, and showcases the model's ability to extrapolate beyond just length generalization.

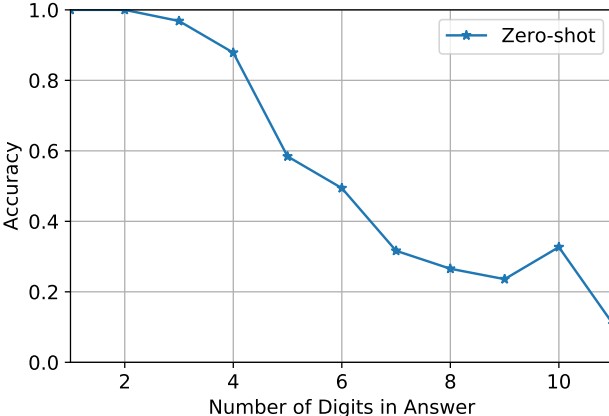

Figure 11: Zero-shot accuracy of one-digit multiplication, where the other number can have up to 11 digits. We see that the accuracy starts to drop after 3 digits in the answer.

**Parity:** Similar to (Anil et al., 2022) we investigate the parity problem as an example of length generalization. We use algorithmic prompting for parity and compare its performance to a few-shot baseline, as well as to a scratchpad-style prompt as discussed in Anil et al. (2022). Figure 12 captures the performance of these three approaches on lists of varying sizes. We use 2 shots of up to 8-digits in answer length in the prompt. Each point in Figure 12 represents average over 100 random samples and we use the same examples for all methods. We observe that algorithmic prompt significantly outperforms both baselines and while the baselines' performance reaches random chance (50%) around length 5, algorithmic prompt keeps accuracy of around 80% to lists of up to 30 digits. Section B.5 and B.6 depict the prompts used in this experiment.

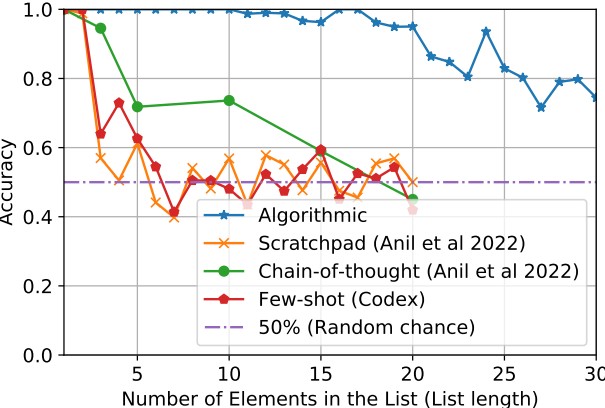

Figure 12: Investigating the performance of algorithmic prompt on parity problem and comparing it to scratchpad few-shot prompt of Anil et al. (2022) as well as few-shot prompting OpenAI's Codex. Each point on the Algorithmic plot corresponds to 100 random samples of a binary list of the same length. Sections B.5, B.6 depict the prompts used for algorithmic and scratchpad methods. The number of examples used in the prompt is two. The scratchpad method uses the prompt from Figure 9 in (Anil et al., 2022). Chain-of-thought values are directly copied from Figure 7 in Anil et al. (2022) (few shot finetuning - few shot eval), and correspond to prompt from Figure 10 in Anil et al. (2022).

## A.5 ADDITIONAL RESULTS FOR SKILL ACCUMULATION

This section includes additional details and figures on skill accumulation from Section 4.

We study whether the superior performance of the algorithmic prompt can be attributed to the fact that it is much longer than the few-shot prompt. To control for this variation, we perform an ablation on the addition-subtraction of the few-shot baseline. We generate $n$ examples of addition and sub-

traction, such that the total number of tokens is equal to the number of tokens used in the algorithmic prompt. Figure 13 shows that having more few-shot examples does not improve performance.

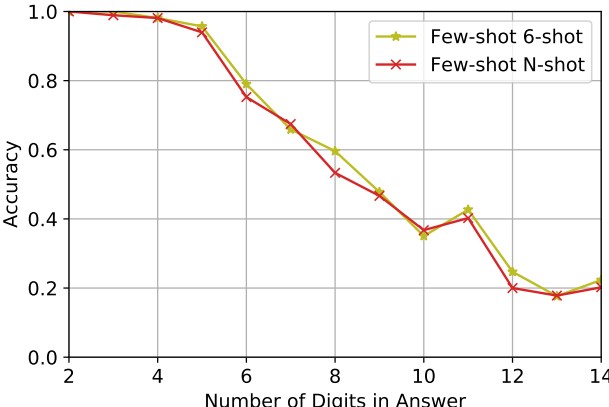

Figure 13: Accuracy on combined addition-subtraction questions using a few-shot prompt. Since the algorithmic prompt uses more tokens in the prompt, we perform an ablation for the few-shot baseline and use $n$ number of examples in the prompt, where $n$ is chosen such that the prompt length matches the algorithmic prompt. We see that there is no improvements in performance beyond the 6 examples used in the baseline.

To further study the effects of teaching addition alongside subtraction, we evaluate two subtraction-only prompts. The first one removes the addition-only prompt examples from the combined addition-subtraction prompt. In the combined prompt, 6 examples are provided, with 2 of them being addition-only examples. After removing the addition-only examples, we are left with 4 subtraction-only examples in the prompt. The second subtraction-only prompt matches the number of shots as the original combined prompt, but includes only subtraction-only examples for all 6 shots. The results are shown in Figure 14(a). We see that using only the 4 subtraction-only prompt examples (*Combined Algo, Sub examples-only*) results in a significant decrease in performance compared to the combined algorithmic prompt. However, when we are able to match the same number of shots (6) as the combined prompt (*Sub-only Algo*), we can recover the original performance. This demonstrates the synergy and positive transfer when simultaneously learning algorithms that share similarities. As a control experiment, we also observe in Figure 14(b) that adding more shots to an addition-only prompt does not improve performance beyond the original prompt, which supports the conclusion that addition-only performance using the combined prompt is not harmed by having other algorithms in the same prompt.

Finally, we note that the prompt development for this task was non-trivial, and the best performance required adding all combinations of positive and negative numbers in the prompt. Thus, scaling to larger number of algorithms may call for more efficient strategies.

### A.6    ADDITIONAL RESULTS FOR SKILL COMPOSITION

This section includes additional details and figures on skill composition from Section 5. In Figure 15, we provide an illustrative demonstration of the change in the prompt when going from two-number addition to multi-number addition to multiplication-as-addition, showing the progression in complexity.

In Figure 17, we include the results for the entire evaluation dataset, including examples that ran out of context in the first pass through the model. In Figure 18, we show the same results but using the count of numbers being added as the x-axis. We employ a second-pass strategy, where we append the last completed step from the first-pass output to the original test question, and perform another inference pass using the new prompt. We observe that this simple second-pass strategy allows us to correctly solve a portion of the questions that were previously incomplete. However, the performance is still significantly below the hypothetical upper estimate performance achieved by first-pass completed questions.

In Figure 19, we use a dialogue-like approach where we employ two models loaded with specialized prompts. For multi-number addition, we prompt one model with an example that explains how to

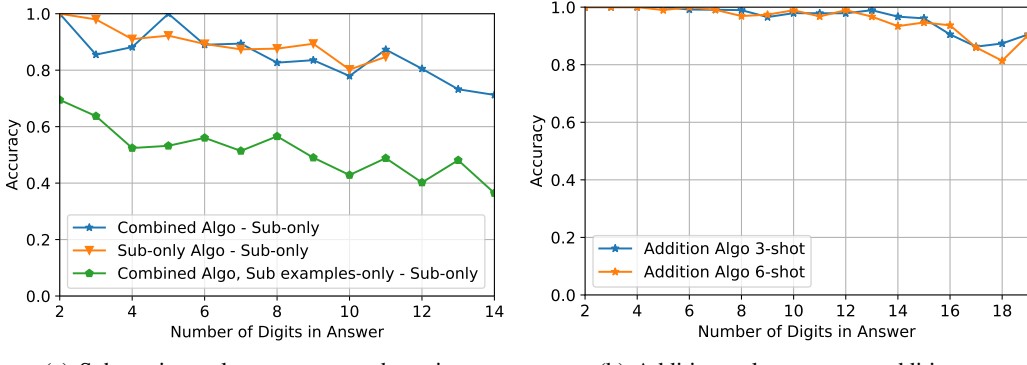

(a) Subtraction-only prompts on subtraction    (b) Addition-only prompt on addition

Figure 14: **Left:** Performance on subtraction-only questions. "Combined Algo" refers to using 4 examples of subtraction questions and 2 examples of addition questions within the prompt. "Sub-only Algo" refers to using 6 examples of subtraction questions within the prompt. "Combined Algo, Sub examples-only" refers to using 4 examples of subtraction questions within the prompt. We see that removing addition-only examples from the prompt significantly harms performance on subtraction-only questions, thus showing the positive transfer that comes from having addition examples. **Right:** Performance on addition-only questions using addition-only prompts with different number of shots. We see that performance of the original addition prompt on the addition task is already saturated on the number of shots.

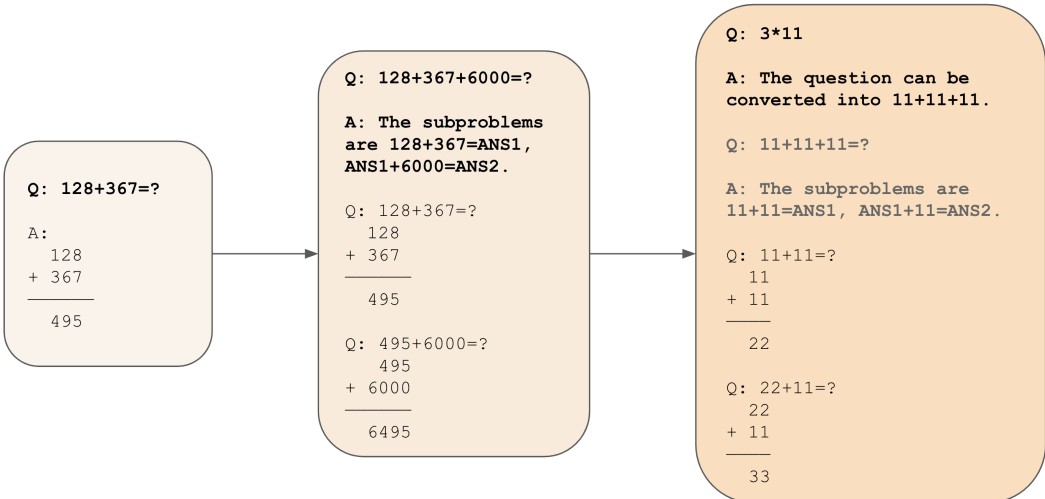

Figure 15: Illustration of the tasks and algorithmic prompting strategies considered for skill composition in Section 5. The actual prompts use algorithmic prompting for each addition question. Starting from simple two-number addition, we explore multi-number addition which decomposes the problem into a set of two-number addition questions, then extend it further to multiplication-as-addition which converts a multiplication question into an equivalent multi-number addition question.

solve multi-number addition problems as a sequence of two-number addition problems, and prompt a second model with the algorithmic prompt for two-number addition. Within the prompt for multi-number addition, we employed specialized tokens to indicate the start and end of a two-number addition problem that the model needs to query the addition-prompted model for. We extract the two-number addition question and send it to the second model, then retrieve the answer and allow the first model to continue with its output. We use the same strategy for multiplication-as-addition. The prompt of this first model can be found in Section B.10 for multi-number addition, and Section B.11 for multiplication-as-addition. We find in Figure 19 that we are able to generalize out-of-distribution from a single prompt example, and avoid context length limitations when evaluated on the longest problems in the evaluation data. This approach allows us to achieve performance comparable to those in Figure 6 on the full dataset.

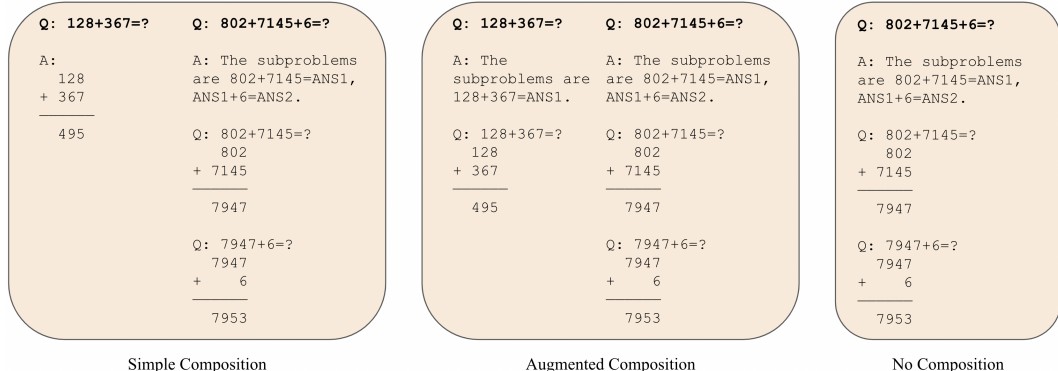

Figure 16: Illustration of various composition strategies. The actual prompts use algorithmic prompting for each addition question. Simple composition combines the prompt from a previously taught skill to new examples illustrating the composed skill. Augmented composition changes the previous prompt examples so that they are treated as special cases of the new composed skill. No composition includes only examples illustrating the new composed skill.

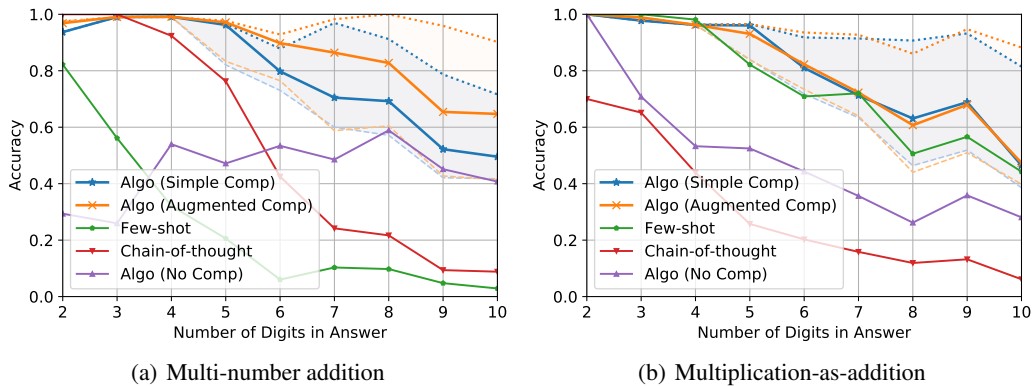

(a) Multi-number addition

(b) Multiplication-as-addition

Figure 17: Performance on compositions of skills. Due to length of the algorithmic output for this task, a number of the longest examples exceed the context length limit for Codex. We employ a second pass strategy to get a final answer for the incomplete questions, where we keep in-context only the last completed state from the first pass. The dotted lines consider only questions for which the model completes the output within one pass, and provide an upper estimate on performance. The dashed lines consider all incomplete questions as false, and provide an lower estimate on performance. We observe that although the algorithmic prompting methods are suffering from having to do a second pass for the longer samples in this task, they still demonstrate better generalization than the baseline methods. Note that for multiplication, we evaluate the algorithmic methods on a harder task than the few-shot baselines, since we force the model to convert the question into the addition of a number $n$ times, while for other baselines we simply perform $1 \times n$-digit multiplication directly.

**Can the model extrapolate to novel compositions?** We explore whether the model can generalize to a new setting than what is demonstrated in the composition skills. In Figure 19 we explored learning multi-number addition by calling the addition algorithmic prompt for each two-number addition subroutine. Now, we evaluate the same multi-number addition prompt to see if it is able to solve multi-number addition-subtraction questions (an extension to the setting in Section 4. We test the model's generalization ability for composition since the model does not see any example of multi-number subtraction. Here, the only change is that we call on the addition-subtraction combined prompt each time the specialized tokens are outputted. The results are shown in Figure 20. We see that the model is able to perform multi-number addition-subtraction despite never seeing a demonstration of this skill. The algorithmic prompt outperforms the few-shot and chain-of-thought baselines while demonstrating good length generalization.

## A.7 ADDITIONAL RESULTS FOR TOOL USE

This section includes additional details and figures for tool use in Section 6.

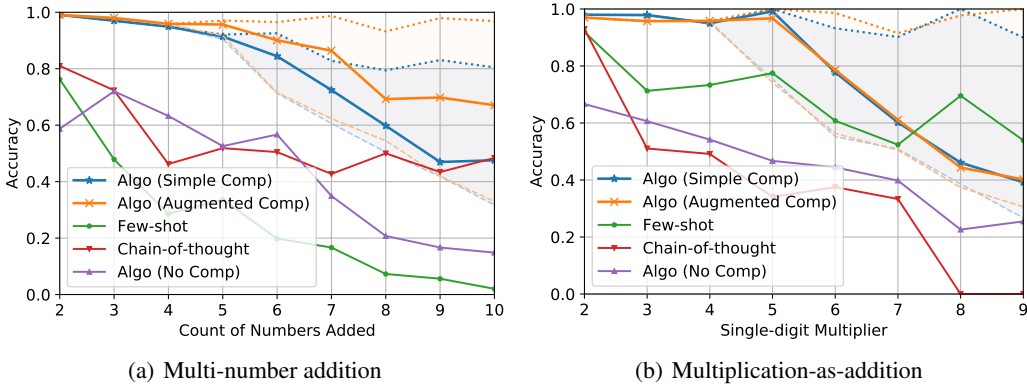

(a) Multi-number addition

(b) Multiplication-as-addition

Figure 18: Performance on compositions of skills with the x-axis being the count of numbers being added instead of the number of digits in answer used in Figure 17. Due to length of the algorithmic output for this task, a number of the longest examples exceed the context length limit for Codex. We employ a second pass strategy to get a final answer for the incomplete questions, where we keep in-context only the last completed state from the first pass. The dotted lines consider only questions for which the model completes the output within one pass, and provide an upper estimate on performance. The dashed lines consider all incomplete questions as false, and provide an lower estimate on performance. We observe that although the algorithmic prompting methods are suffering from having to do a second pass for the longer samples in this task, they still demonstrate better generalization than the baseline methods. Note that for multiplication, we evaluate the algorithmic methods on a harder task than the few-shot baselines, since we force the model to convert the question into the addition of a number $n$ times, while for other baselines we simply perform $1 \times n$-digit multiplication directly.

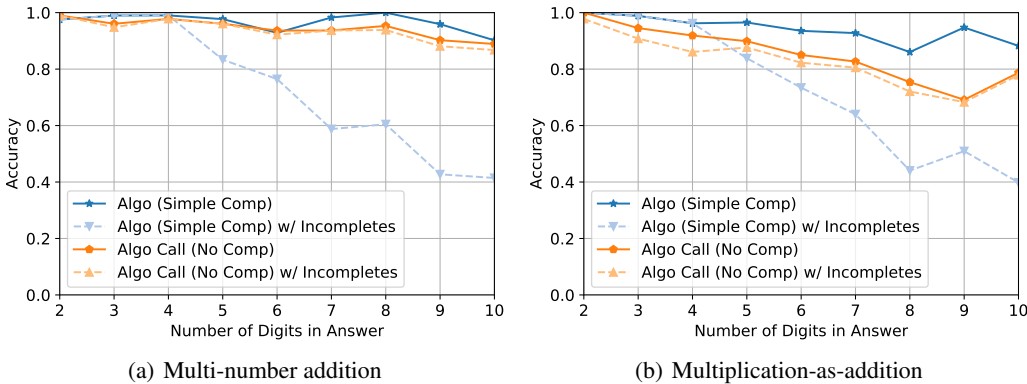

(a) Multi-number addition

(b) Multiplication-as-addition

Figure 19: Performance on compositions of skills with algorithmic calls. Due to length of the algorithmic output for this task, a number of the longest examples exceed the context length limit for Codex. We employ a dialogue-like strategy to get a final answer for the incomplete questions, where we allow models loaded with different prompts to interact with each other through the use of specialized tokens learned in-context. The dashed lines consider all incomplete questions as false, and provide an lower estimate on performance. "Algo Call" refers to this dialogue-like method, which is akin to the "No Composition" setup since we do not include the two-number addition examples in the prompt. We find that we are able to generalize out-of-distribution from a single prompt example of digits length 2, and avoid context length limitations when evaluated on the longest problems in the evaluation data.

**Dataset:** Figure 21 shows an example question and answer pair from GSM8k with chain-of-thought rationale. For the GSM8k-Hard dataset, we increase the numerical values used in the questions, thus making the task more difficult for the model. The number of digits in the answer range from 3 to 12, with an average length of 7.2. In the original GSM8k addition-only subset, the number of digits range from 1 to 5 with an average length of 2.4. An example is presented in Figure 7(b).

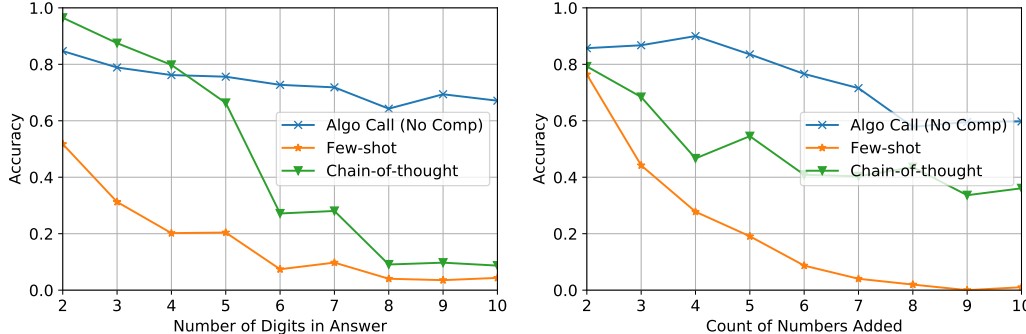

Figure 20: Performance on multi-number addition-subtraction questions with algorithmic calls. This dataset consists of 1000 questions of multi-number addition where each number could be positive or negative. We treat all incomplete questions as wrong. We use the same composition prompt as Figure 19(a), which only demonstrates an example of multi-number addition. In this case, we call on the addition-subtraction algorithm each time the specialized tokens are called. We see that the algorithmic call method outperforms the chain-of-thought and few-shot baselines and demonstrates length generalization. **Left:** Accuracy with number of digits in the final answer on the x-axis. **Right:** Accuracy with count of numbers in the equation on the x-axis.

> **Q:** Tommy has 3 toy cars. His neighbor, Jessie, has 3 cars too. Jessie's older brother has 5 more cars than Tommy and Jessie. How many cars do the three of them have altogether?
> **A:** Tommy and Jessie have 3+3=6 cars. Jessie's brother has 5+6=11 cars. Altogether, they have 6+11=17 cars. The answer is 17.

Figure 21: An example question and answer pair from GSM8k with chain-of-thought rationale.

### A.7.1 ADDITIONAL DETAILS ON AUGMENTING INFORMAL MATHEMATICAL REASONING

There are two challenges in augmenting algorithmic prompt to the chain-of-thought prompt: 1) since there are many instances of addition in the chain-of-thought examples, this prompt would take up a large number of tokens, and 2) we have seen previously that combining similar skills like addition and subtraction within the same prompt did not result in any interference (with evidence of positive transfer), but since informal mathematical reasoning and arithmetic operations are very different skills, this may no longer be the case.

To address the first challenge (lengthy prompt), we only embed the addition algorithm in 2 of the prompt examples, and we indicate these augmented examples through the <ALGO> flag while the remaining 6 examples use the <NONALGO> flag. These flags allow us to control whether the model should perform addition using the algorithm or by direct calculation. Thus, for each setting, we run two experiments by appending the <ALGO> or <NONALGO> flag to the test question. This flag-based strategy is simple yet effective, with $86\%$ of <ALGO> examples and $0\%$ of <NONALGO> examples exhibiting algorithmic output. See Section B.13 for the actual prompt.

For the second challenge (interference), we hypothesize that explicitly presenting the summary of the solution may help to disentangle the two skills (i.e. informal mathematical reasoning and arithmetic operation). Thus we explore a version of chain-of-thought where the answer begins with an overall plan/summary of the solution steps, before the individual steps are explained. We refer to this version as "with plan", and refer to the baseline version without a summary as "no plan". The actual prompt is shown in Section B.13.

Figure 7(a) shows the results using this approach. First, we evaluate the impact of including algorithmic output in the prompt by comparing the chain-of-thought baseline with ("no plan no algo") and without ("no plan w/ algo") algorithmic output for addition questions. We find that including algorithmic output in the examples significantly disrupts the model's informal mathematical reasoning abilities in the <ALGO> experiment, but leaves the <NONALGO> performance relatively unchanged. This demonstrates the existence of interference between the two skills. We conjecture that this occurs when we mix highly different skills within the same context. The informal mathematical reasoning component relies on the model's pretraining knowledge, while the algorithmic component is regimented and requires the model to follow specific instructions, and the different

nature and format of these two skills appears to interfere with their performance. Next, we evaluate the impact of having a solution plan at the beginning of the output. Comparing the performance of "w/ plan w/ algo" and "no plan w/ algo", we see that the solution plan alleviates some of the interference seen in the <ALGO> experiment. Nonetheless, the performance is still much worse than the same version without algorithmic output ("w/ plan no algo"). In summary, we identify an interference phenomenon which may occur when combining skills of different kind within the same context, and find that using flags in the prompt can be a simple way of directing a model's attention as <NONALGO> experiments do not suffer from interference the way that <ALGO> experiments do.

### A.7.2 ADDITIONAL DETAILS ON ALGORITHMIC PROMPT AS TOOL USE

In this approach, we utilize one model for performing the informal mathematical reasoning steps and a separate model for doing algorithmic addition calculations. To enable a dialogue-like interaction, we teach the first model to output specific tokens to indicate when a separate model should be consulted. See Figure 7(b) for an example of how these tokens are used. We then extract the addition question using these tokens and send it to the second model loaded with the addition algorithmic prompt, which executes the addition algorithm and returns the answer back to the first model. The first model would then continue with the rest of the answer without needing to keep the algorithmic output in its context. Creswell and Shanahan (2022) uses a similar multi-model and multi-prompt strategy in order to separate out selection from inference in reasoning problems. This approach can be considered a form of tool use (Parisi et al., 2022), where a model queries another source for a particular type of information.

### A.8 OTHER RESULTS ON TWO-NUMBER ADDITION

**Chain-of-thought prompt comparison:** In order to generate chain-of-thought prompt templates using the "let's think step-by-step" approach, we first set the temperature to 0.3 and generate a number of samples. We find that the output of the model centers around two formats, shown in Section B.1.3. We convert both formats into a chain-of-thought prompt and evaluate their performance. We choose the better performing version as the baseline in the main results.

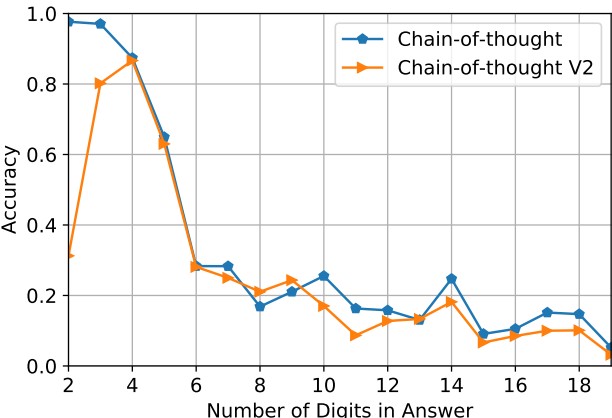

Figure 22: Performance of two-number addition with two versions of chain-of-thought prompt. Due to the stochasticity of the "let's think step-by-step" approach, there can be multiple strategies being generated by the model. "Chain-of-thought" is the one that we use in the main results. "Chain-of-thought V2" is an alternative version shown in Section B.1.3. We choose the better performing version as the main chain-of-thought method.

**Other baselines:** There have been other attempts of understanding and developing improved versions of scratchpad and chain-of-thought prompts for addition. For example, Akyurek and Akyurek (2022) provides an analysis of the scratchpad approach to prompting. They studied various style choices and natural language descriptions and evaluated their impact on performance. Moreover, Zhou (2022) and Chen (2022) showed new chain-of-thought approaches to addition that is different than the one we find from using "let's think step by step." We study these approaches under our evaluation setting, and the results are shown in Figure 23. We find that the performance of these methods are lower than existing baselines. For these methods, we adopt the same 3 prompt exam-

ples that we use for all methods. We note that our evaluation dataset allows for the possibility that the two number have different lengths, which is different from the results shown in these works that use the same length for both numbers. Moreover, for these methods, the questions are represented in delimited format (e.g. 1,2,8), while all the other methods use standard number representations (e.g. 128). We have observed from our scratchpad experiments that having delimiters tend to help scratchpad-like prompts and hurt few-shot methods.

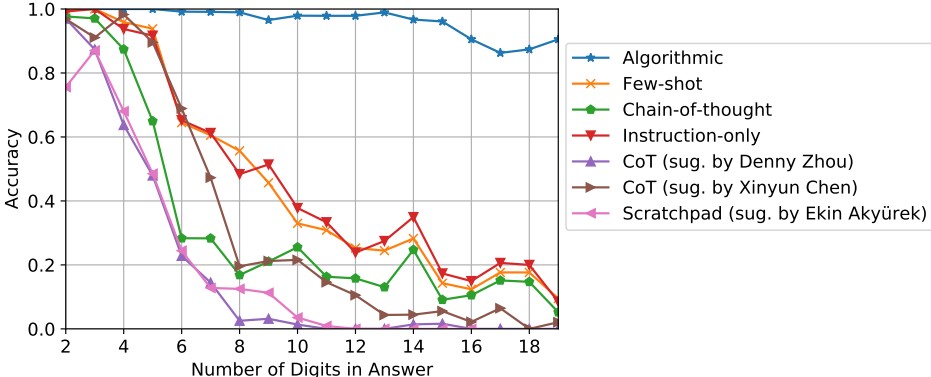

Figure 23: Performance of two-number addition with additional chain-of-thought ("CoT") and scratchpad baselines. We find that these methods perform worse than existing baselines and do not demonstrate significant length generalization. Crucially, although these prompts are augmented with more natural language descriptions, they do not use unambiguous rules like algorithmic prompt does, which explains the performance gap.

Moreover, although Akyurek and Akyurek (2022) also cite the importance of natural language descriptions in the prompt and not leaving things up for interpretation, their conclusion is based on having markers for different values and standard formats for the carry component. Although our prompts also use natural language descriptions, we do so mainly to explain more abstract concepts that are harder to illustrate, such as the grouping operation in memorized multiplication (see Section 3.2). We have shown in Figure 9 that a symbols-only addition prompt with minimal natural language still performs significantly better than all baselines and achieves length generalization. In addition, we provide explicit explanation in the sense of giving the actual equations for deriving certain values and replacing uncommon operations with more common ones, which is the key contributor to the high performance of algorithmic prompts. In fact, we find that having inconsistent markers for carry can still significantly outperform a consistent template when more explicit rules are provided, contrary to the claims in Akyurek and Akyurek (2022). In Figure 24, we show that using a prompt with *inconsistent* carry formats (i.e. carry is only included in the equation if it is 1, and omitted when it is 0) but with the inclusion of the modulo operator to extract the answer digits significantly outperforms the version with *consistent* carry formats. Here, we compare to the algorithmic prompting with nonexplicit calculation ablation from Figure 3(a), and both prompts do not include the effect of explicit carry equations. We also use an inconsistent format for the modulo operator, only displaying this step when the digit sum is greater than 9. This shows that having more explicit rules is much more crucial than having standardized formats in the prompt. The prompts can be found in Section B.2.2. Overall, these results differentiate the insights in this work with those in Akyurek and Akyurek (2022).

# B    PROMPT EXAMPLES

## B.1    ADDITION PROMPT STRATEGIES

For addition prompts, we use 3-shot with the examples $128+367$, $9980+29$, and $802+7145$ in order. For conciseness, we may include only subsets of the prompt questions in the prompt examples.

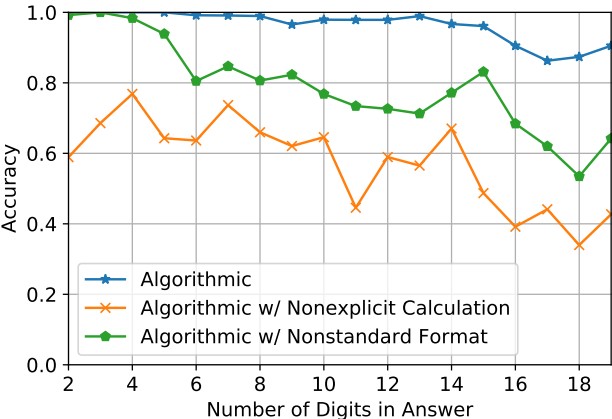

Figure 24: Performance of two-number addition with variants of algorithmic prompt. Both "algorithmic w/ nonexplicit calculation" and "algorithmic w/ nonstandard format" do not contain the equation for the carry value derivations. However, "algorithmic w/ nonstandard format" further do not use a standard format for the carry values, omitting it completely when the value is 0. "Algorithmic w/ nonstandard format" further includes the modulo operation for deriving the answer digit, again in nonstandard format. Prompts can be found in Section B.2.2. We find that "algorithmic w/ nonstandard format" significantly outperforms the one with standard format, illustrating that having more explicit rules is much more crucial than having standardized formats in the prompt.

### B.1.1 ALGORITHMIC PROMPT FOR ADDITION

---

**Problem: 128+367=**
Explanation:
The first number is 128, FN=[1,2,8]. The second number is 367,
SN=[3,6,7]. Since FN [1,2,8] has 3 digits, SN [3,6,7] has 3 digits, thus
the maximum number of digits is 3. In each subsequent step, we remove
one number from the end of FN and one from the end of SN.
Length of FN is 3. FN=[1,2,8]. Length of SN is 3. SN=[3,6,7].
FN[3]=8. SN[3]=7. C[3]=0. Since 8+7+0=15, 15>10, 15%10=5. Length of A
is 1. Thus A=[5]. Since (15-5)/10=1, C[2]=1.
Length of FN is 2. FN=[1,2]. Length of SN is 2. SN=[3,6]. FN[2]=2.
SN[2]=6. C[2]=1. Since 2+6+1=9, 9<10, 9%10=9. Length of A is 2. Thus
A=[9,5]. Since (9-9)/10=0, C[1]=0.
Length of FN is 1. FN=[1]. Length of SN is 1. SN=[3]. FN[1]=1.
SN[1]=3. C[1]=0. Since 1+3+0=4, 4<10, 4%10=4. Length of A is 3. Thus
A=[4,9,5]. Since (4-4)/10=0, C[0]=0.
There are no more digits and C[0]=0. Thus the process is complete.
Since there are no more operators, the problem is complete. The final
Answer is [4,9,5].
**Problem: 9980+29=**
Explanation:
The first number is 9980, FN=[9,9,8,0]. The second number is 29,
SN=[2,9]. Since FN [9,9,8,0] has 4 digits, SN [2,9] has 2 digits, thus
the maximum number of digits is 4. In each subsequent step, we remove
one number from the end of FN and one from the end of SN.
Length of FN is 4. FN=[9,9,8,0]. Length of SN is 2. SN=[2,9].
FN[4]=0. SN[4]=9. C[4]=0. Since 0+9+0=9, 9<10, 9%10=9. Length of A
is 1. Thus A=[9]. Since (9-9)/10=0, C[3]=0.
Length of FN is 3. FN=[9,9,8]. Length of SN is 1. SN=[2]. FN[3]=8.
SN[3]=2. C[3]=0. Since 8+2+0=10, 10=10, 10%10=0. Length of A is 2.
Thus A=[0,9]. Since (10-0)/10=1, C[2]=1.
Length of FN is 2. FN=[9,9]. Length of SN is 0. SN=[]. FN[2]=9. SN
is empty. C[2]=1. Since 9+0+1=10, 10=10, 10%10=0. Length of A is 3.
Thus A=[0,0,9]. Since (10-0)/10=1, C[1]=1.
Length of FN is 1. FN=[9]. Length of SN is 0. SN=[]. FN[1]=9. SN is
empty. C[1]=1. Since 9+0+1=10, 10=10, 10%10=0. Length of A is 4. Thus
A=[0,0,0,9]. Since (10-0)/10=1, C[0]=1.
There are no more digits, but C[0]=1. Length of A is 5. Thus
A=[1,0,0,0,9].
There are no more digits and the process is complete. Since there
are no more operators, the problem is complete. The final Answer is
[1,0,0,0,9].
**Problem: 802+7145=**
Explanation:
The first number is 802, FN=[8,0,2]. The second number is 7145,
SN=[7,1,4,5]. Since FN=[8,0,2] has 3 digits, SN=[7,1,4,5] has 4 digits,
thus the maximum number of digits is 4. In each subsequent step, we
remove one number from the end of FN and one from the end of SN.
Length of FN is 3. FN=[8,0,2]. Length of SN is 4. SN=[7,1,4,5].
FN[4]=2. SN[4]=5. C[4]=0. Since 2+5+0=7, 7<10, 7%10=7. Length of A
is 1. Thus A=[7]. Since (7-7)/10=0, C[3]=0.
Length of FN is 2. FN=[8,0]. Length of SN is 3. SN=[7,1,4]. FN[3]=0.
SN[3]=4. C[3]=0. Since 0+4+0=4, 4<10, 4%10=4. Length of A is 2. Thus
A=[4,7]. Since (4-4)/10=0, C[2]=0.
Length of FN is 1. FN=[8]. Length of SN is 2. SN=[7,1]. FN[2]=8.
SN[2]=1. C[2]=0. Since 8+1+0=9, 9<10, 9%10=9. Length of A is 3. Thus
A=[9,4,7]. Since (9-9)/10=0, C[1]=0.
Length of FN is 0. FN=[]. Length of SN is 1. SN=[7]. FN is empty.
SN[1]=7. C[1]=0. Since 0+7+0=7, 7<10, 7%10=7. Length of A is 4. Thus
A=[7,9,4,7]. Since (7-7)/10=0, C[0]=0.
There are no more digits and C[0]=0. Thus the process is complete.
Since there are no more operators, the problem is complete. The final
Answer is [7,9,4,7].

---

### B.1.2 FEW-SHOT PROMPT FOR ADDITION

```
Q: 128+367=
A: 495.
Q: 9980+29=
A: 10009.
Q: 802+7145=
A: 7947.
```

### B.1.3 CHAIN-OF-THOUGHT PROMPT FOR ADDITION

```
Problem: 128+367=?
Explanation:  Let's think step by step.
128+367=128+300+67=428+67=495.  The final Answer is 495.

Problem: 9980+29=?
Explanation:  Let's think step by step.
9980+29=9980+20+9=10000+9=10009.  The final Answer is 10009.

Problem: 802+7145=?
Explanation:  Let's think step by step.
802+7145=802+7000+100+45=7802+100+45=7902+45=7947.  The final Answer is
7947.
```

The following is another version of chain-of-thought prompt that we tried, but the performance was worse than the previous prompt.

```
Problem: 128+367=?
Explanation:  Let's think step by step.
1.  128+367=?
2.  128+300=428
3.  428+60=488
4.  488+7=495
5.  495=495
So, 128+367=495.  The final Answer is 495.
```

### B.1.4 INSTRUCTION ADDITION PROMPT FOR ADDITION

```
The following are instructions for solving addition problems in the form
of x + y = z, where x, y, and z are positive integers.
We will use the standard algorithm for addition.  We align the numbers x
and y on the least significant digit, which is the ones digit.  Starting
from right to left, we go from the least significant digit to the most
significant digit and add the corresponding digits from each number.
When the sum of the two digits is greater than 9, a carry of 1 is
included in the sum of the next digits.  When there is only one digit
available from the two numbers, only that digit along with any carry
is included in the sum.  When all the digits are processed, only the
remaining carry if any shall be included in the sum.
For x + y = z where x = int(str(abc)), y = int(str(defg)), we can solve z
with the following steps:
1) c+g=w', w=w'%10
2) b+f+((w'-w)/10)=v', v=v'%10
3) a+e+((v'-v)/10)=u', u=u'%10
4) d+((u'-u)/10)=t', t=t'%10
5) s=(t'-t)/10
Thus, z = int(str(stuvw)).
The answer should be in the form below:
Q: What is abc+defg=?
A: abc
+defg
-------
stuvw
The answer is stuvw.
```

### B.1.5 SCRATCHPAD PROMPT FOR ADDITION

```
Input:
128+367
Target:
<scratch>
1 2 8 + 3 6 7 , C: 0
1 2 + 3 6 , 5 C: 1
1 + 3 , 9 5 C: 0
, 4 9 5 C: 0
4 9 5
</scratch>4 9 5.

Input:
9980+29
Target:
<scratch>
9 9 8 0 + 2 9 , C: 0
9 9 8 + 2 , 9 C: 0
9 9 , 0 9 C: 1
9 , 0 0 9 C: 1
, 0 0 0 9 C: 1
1 0 0 0 9
</scratch>1 0 0 0 9.
```

### B.1.6 DETAILED SCRATCHPAD PROMPT FOR ADDITION

```
Input:
128+367
Target:
<scratch>
1 2 8 has 3 digits.
3 6 7 has 3 digits.
1 2 8 + 3 6 7 , C=0 , 8 + 7 + 0 = 1 5 , A->5 , C->1
1 2 + 3 6 , A=5 , C=1 , 2 + 6 + 1 = 9 , A->9 , C->0
1 + 3 , A=9 5 , C=0 , 1 + 3 + 0 = 4 , A->4 , C->0
+ , A=4 9 5 , C=0 , END
</scratch>
4 9 5

Input:
9980+29
Target:
<scratch>
9 9 8 0 has 4 digits.
2 9 has 2 digits.
9 9 8 0 + 2 9 , C=0 , 0 + 9 + 0 = 9 , A->9 , C->0
9 9 8 + 2 , A=9 , C=0 , 8 + 2 + 0 = 1 0 , A->0 , C->1
9 9 + , A=0 9 , C=1 , 9 + 0 + 1 = 1 0 , A->0 , C->1
9 + , A=0 0 9 , C=1 , 9 + 0 + 1 = 1 0 , A->0 , C->1
+ , A=0 0 0 9 , C=1 , 0 + 0 + 1 = 1 , A->1 , C->0
+ , A=1 0 0 0 9 , C=0 , END
</scratch>
1 0 0 0 9
```

## B.2 ALGORITHMIC PROMPT ABLATIONS FOR ADDITION

### B.2.1 ALGORITHMIC PROMPT WITH UNCOMMON OPERATIONS FOR ADDITION

The uncommon indexing operation is highlighted in red.

```
Problem: 128+367=
Explanation:
The first number is 128, FN=[1,2,8].  The second number is 367,
SN=[3,6,7].  Since FN [1,2,8] has 3 digits, SN [3,6,7] has 3 digits, thus
the maximum number of digits is 3.  In each subsequent step, we remove
one number from the end of FN and one from the end of SN.
FN[3]=8.  SN[3]=7.  C[3]=0.  Since 8+7+0=15, 15>10, 15%10=5.  Length of A
is 1.  Thus A=[5].  Since (15-5)/10=1, C[2]=1.
FN[2]=2.  SN[2]=6.  C[2]=1.  Since 2+6+1=9, 9<10, 9%10=9.  Length of A is
2.  Thus A=[9,5].  Since (9-9)/10=0, C[1]=0.
FN[1]=1.  SN[1]=3.  C[1]=0.  Since 1+3+0=4, 4<10, 4%10=4.  Length of A is
3.  Thus A=[4,9,5].  Since (4-4)/10=0, C[0]=0.
There are no more digits and C[0]=0.  Thus the process is complete.
Since there are no more operators, the problem is complete.  The final
Answer is [4,9,5].
```

### B.2.2   ALGORITHMIC PROMPT WITH NON-EXPLICIT CARRY FOR ADDITION

In this prompt, the explicit carry calculations in the prompt in Section B.1.1 is omitted.

```
Problem: 128+367=
Explanation:
The first number is 128, FN=[1,2,8].  The second number is 367,
SN=[3,6,7].  Since FN [1,2,8] has 3 digits, SN [3,6,7] has 3 digits, thus
the maximum number of digits is 3.  In each subsequent step, we remove
one number from the end of FN and one from the end of SN.
Length of FN is 3.  FN=[1,2,8].  Length of SN is 3.  SN=[3,6,7].
FN[3]=8.  SN[3]=7.  C[3]=0.  Since 8+7+0=15.  Length of A is 1.  Thus
A=[5].  C[2]=1.
Length of FN is 2.  FN=[1,2].  Length of SN is 2.  SN=[3,6].  FN[2]=2.
SN[2]=6.  C[2]=1.  Since 2+6+1=9.  Length of A is 2.  Thus A=[9,5].
C[1]=0.
Length of FN is 1.  FN=[1].  Length of SN is 1.  SN=[3].  FN[1]=1.
SN[1]=3.  C[1]=0.  Since 1+3+0=4.  Length of A is 3.  Thus A=[4,9,5].
C[0]=0.
There are no more digits and C[0]=0.  Thus the process is complete.
Since there are no more operators, the problem is complete.  The final
Answer is [4,9,5].
```

In this prompt, the explicit carry calculations in the prompt in Section B.1.1 is omitted. In addition, the carry format is inconsistent, omitting the carry completely when the value is 0. This prompt is used for the ablation in Figure 24.

```
Problem: 128+367=
Explanation:
The first number is 128, FN=[1,2,8].  The second number is 367,
SN=[3,6,7].  Since FN [1,2,8] has 3 digits, SN [3,6,7] has 3 digits, thus
the maximum number of digits is 3.  In each subsequent step, we remove
one number from the end of FN and one from the end of SN.
Length of FN is 3.  FN=[1,2,8].  Length of SN is 3.  SN=[3,6,7].
FN[3]=8.  SN[3]=7.  C[3]=0.  Since 8+7=15, 15>10, 15%10=5.  Length of
A is 1.  Thus A=[5].  C[2]=1.  Length of FN is 2.  FN=[1,2].  Length of
SN is 2.
SN=[3,6].  FN[2]=2.  SN[2]=6.  C[2]=1.  Since 2+6+1=9, 9<10.  Length of A
is 2.  Thus A=[9,5].  C[1]=0.
Length of FN is 1.  FN=[1].  Length of SN is 1.  SN=[3].  FN[1]=1.
SN[1]=3.  C[1]=0.  Since 1+3=4, 4<10.  Length of A is 3.  Thus A=[4,9,5].
C[0]=0.
There are no more digits and C[0]=0.  Thus the process is complete.
Since there are no more operators, the problem is complete.  The final
Answer is [4,9,5].
```

### B.2.3 ALGORITHMIC PROMPT FOR ADDITION WITH IRREGULAR ERRORS

The errors are highlighted in red.

```
Problem: 128+367=
Explanation:
The first number is 128, FN=[1,2,8].  The second number is 367,
SN=[3,6,7].  Since FN [1,2,8] has 3 digits, SN [3,6,7] has 3 digits, thus
the maximum number of digits is 3.  In each subsequent step, we remove
one number from the end of FN and one from the end of SN.
Length of FN is 3.  FN=[1,2,8].  Length of SN is 3.  SN=[3,6,7].
FN[3]=8.  SN[3]=7.  C[3]=0.  Since 8+6+0=15, 15>10, 15%10=5.  Length of A
is 1.  Thus A=[5].  Since (15-5)/10=1, C[2]=1.
Length of FN is 2.  FN=[1,2].  Length of SN is 2.  SN=[3,6].  FN[2]=2.
SN[2]=6.  C[2]=1.  Since 2+6+1=9, 9<10, 9%10=9.  Length of A is 2.  Thus
A=[9,5].  Since (9-9)/10=0, C[1]=0.
Length of FN is 1.  FN=[1].  Length of SN is 1.  SN=[3].  FN[1]=1.
SN[1]=3.  C[1]=0.  Since 1+2+0=4, 4<10, 4%10=4.  Length of A is 3.  Thus
A=[4,9,5].  Since (4-4)/10=0, C[0]=0.
There are no more digits and C[0]=0.  Thus the process is complete.
Since there are no more operators, the problem is complete.  The final
Answer is [4,9,5].
```

### B.2.4 ALGORITHMIC PROMPT FOR ADDITION WITH SYSTEMATIC ERRORS

The errors are highlighted in red.

```
Problem: 128+367=
Explanation:
The first number is 128, FN=[1,2,8].  The second number is 367,
SN=[3,6,7].  Since FN [1,2,8] has 3 digits, SN [3,6,7] has 3 digits, thus
the maximum number of digits is 3.  In each subsequent step, we remove
one number from the end of FN and one from the end of SN.
Length of FN is 3.  FN=[1,2,8].  Length of SN is 3.  SN=[3,6,7].
FN[3]=8.  SN[3]=7.  C[3]=0.  Since 8+6+0=15, 15>10, 15%10=5.  Length of A
is 1.  Thus A=[5].  Since (15-5)/10=1, C[2]=1.
Length of FN is 2.  FN=[1,2].  Length of SN is 2.  SN=[3,6].  FN[2]=2.
SN[2]=6.  C[2]=1.  Since 2+5+1=9, 9<10, 9%10=9.  Length of A is 2.  Thus
A=[9,5].  Since (9-9)/10=0, C[1]=0.
Length of FN is 1.  FN=[1].  Length of SN is 1.  SN=[3].  FN[1]=1.
SN[1]=3.  C[1]=0.  Since 1+2+0=4, 4<10, 4%10=4.  Length of A is 3.  Thus
A=[4,9,5].  Since (4-4)/10=0, C[0]=0.
There are no more digits and C[0]=0.  Thus the process is complete.
Since there are no more operators, the problem is complete.  The final
Answer is [4,9,5].
```

### B.2.5 SYMBOLS-ONLY ALGORITHMIC PROMPT FOR ADDITION

```
Problem: 128+367=
Explanation:
FN=128, FN=[1,2,8].  SN=367, SN=[3,6,7].  Len(FN)=3, Len(SN)=3, MaxLen=3.
Len(FN)=3.  FN=[1,2,8].  Len(SN)=3.  SN=[3,6,7].  FN[3]=8.  SN[3]=7.
C[3]=0.  8+7+0=15, 15>10, 15%10=5.  Len(A)=1.  A=[5].  (15-5)/10=1,
C[2]=1.
Len(FN)=2.  FN=[1,2].  Len(SN)=2.  SN=[3,6].  FN[2]=2.  SN[2]=6.  C[2]=1.
2+6+1=9, 9<10, 9%10=9.  Len(A)=2.  A=[9,5].  (9-9)/10=0, C[1]=0.
Len(FN)=1.  FN=[1].  Len(SN)=1.  SN=[3].  FN[1]=1.  SN[1]=3.  C[1]=0.
Since 1+3+0=4, 4<10, 4%10=4.  Len(A)=3.  A=[4,9,5].  (4-4)/10=0, C[0]=0.
Len(FN)=0 and Len(SN)=0 and C[0]=0.  Done.  The final Answer is [4,9,5].
```

### B.2.6 SYMBOLS-ONLY ALGORITHMIC PROMPT FOR ADDITION WITHOUT KEYWORDS

In this prompt, we do not use the keywords `Len` and `Max`.

```
Problem: 128+367=
Explanation:
FN=128, FN=[1,2,8]. SN=367, SN=[3,6,7]. VBZ(FN)=3, VBZ(SN)=3, UXOVBZ=3.
VBZ(FN)=3. FN=[1,2,8]. VBZ(SN)=3. SN=[3,6,7]. FN[3]=8. SN[3]=7.
C[3]=0.
8+7+0=15, 15>10, 15%10=5. VBZ(A)=1. A=[5]. (15-5)/10=1, C[2]=1.
VBZ(FN)=2. FN=[1,2]. VBZ(SN)=2. SN=[3,6]. FN[2]=2. SN[2]=6. C[2]=1.
2+6+1=9, 9<10, 9%10=9. VBZ(A)=2. A=[9,5]. (9-9)/10=0, C[1]=0.
VBZ(FN)=1. FN=[1]. VBZ(SN)=1. SN=[3]. FN[1]=1. SN[1]=3. C[1]=0.
Since 1+3+0=4, 4<10, 4%10=4. VBZ(A)=3. A=[4,9,5]. (4-4)/10=0, C[0]=0.
VBZ(FN)=0 and VBZ(SN)=0 and C[0]=0. Done. The final Answer is [4,9,5].
```

### B.2.7 SYMBOLS-ONLY ALGORITHMIC PROMPT FOR ADDITION WITH MISLEADING KEYWORDS

In this prompt, we replace the keywords `Len` and `Max` with `Str` and `Min`.

```
Problem: 128+367=
Explanation:
FN=128, FN=[1,2,8]. SN=367, SN=[3,6,7]. Str(FN)=3, Str(SN)=3, MinStr=3.
Str(FN)=3. FN=[1,2,8]. Str(SN)=3. SN=[3,6,7]. FN[3]=8. SN[3]=7.
C[3]=0. 8+7+0=15, 15>10, 15%10=5. Str(A)=1. A=[5]. (15-5)/10=1,
C[2]=1.
Str(FN)=2. FN=[1,2]. Str(SN)=2. SN=[3,6]. FN[2]=2. SN[2]=6. C[2]=1.
2+6+1=9, 9<10, 9%10=9. Str(A)=2. A=[9,5]. (9-9)/10=0, C[1]=0.
Str(FN)=1. FN=[1]. Str(SN)=1. SN=[3]. FN[1]=1. SN[1]=3. C[1]=0.
Since 1+3+0=4, 4<10, 4%10=4. Str(A)=3. A=[4,9,5]. (4-4)/10=0, C[0]=0.
Str(FN)=0 and Str(SN)=0 and C[0]=0. Done. The final Answer is [4,9,5].
```

## B.3 ADDITION-SUBTRACTION PROMPT STRATEGIES

### B.3.1 ALGORITHMIC PROMPT FOR ADDITION-SUBTRACTION

For the addition-subtraction prompt, we use prompt examples $128 + 367$, $9980 + 29$, $29 - 570$, $-99 - 21$, $483 - 389$, and $-30 + 8002$ in order.

```
Problem: 483-389=
Explanation:
The first number is 483, adding commas between each number, FN=[4,8,3].
The second number is -389, adding commas between each number,
SN=-[3,8,9]. FN [4,8,3] has 3 digits, SN -[3,8,9] has 3 digits, max is
3.
Len(FN)=3. FN=[4,8,3]. FN[3]=3. Len(SN)=3. SN=-[3,8,9]. SN[3]=-9.
C[3]=0. Since 3-9+0=-6, -6<-10, -6%-10=-6. Len(A)=1. A=[-6]. Since
(-6--6)/10=0, C[2]=0.
Len(FN)=2. FN=[4,8]. FN[2]=8. Len(SN)=2. SN=-[3,8]. SN[2]=-8.
C[2]=0. Since 8-8+0=0, 0<10, 0%10=0. Len(A)=2. A=[0,-6]. Since
(0-0)/10=0, C[1]=0.
Len(FN)=1. FN=[4]. FN[1]=4. Len(SN)=1. SN=-[3]. SN[1]=-3. C[1]=0.
Since 4-3+0=1, 1<10, 1%10=1. Len(A)=3. A=[1,0,-6]. Since (1-1)/10=0,
C[0]=0.
Len(FN)=0. FN=[]. FN[0]=empty. Len(SN)=0. SN=-[]. SN[0]=empty.
Since both FN and SN are empty, next. Since C[0]=0, the steps are done.
Since there are - in A, we check the sign of the last step A[1]=1. Since
1 is non-neg, we process A from right to left. A=[1,0,-6]=[+1,+0,-6].
C[3]=0.
Len(A)=3. A=[+1,+0,-6]. A[3]=-6. Since -6<0, B=10, C[2]=-1. Since
C[3]=0, thus -6+10+0=4. Len(ANEW)=1. ANEW=[4]. C[2]=-1.
Len(A)=2. A=[+1,+0]. A[2]=+0. Since +0 is 0, B=0, C[1]=0. Since
C[2]=-1, thus 0+0-1=-1, which is neg, thus repeat with B=10, C[1]=-1.
-1+10+0=9. Len(ANEW)=2. ANEW=[9,4]. C[1]=-1.
###continued on next page
```

```
Len(A)=1.  A=[+1].  A[1]=+1.  Since +1>0, B=0, C[0]=0.  Since C[1]=-1,
thus 1+0-1=0.  Len(ANEW)=3.  ANEW=[0,9,4].  C[0]=0.
Len(A)=0.  A=[].  Since A is empty, the problem is complete.  The final
Answer is [0,9,4].
```
**Problem: 29-570=**
```
Explanation:
The first number is 29, adding commas between each number, FN=[2,9].  The
second number is -570, adding commas between each number, SN=-[5,7,0].
FN [2,9] has 2 digits, SN -[5,7,0] has 3 digits, max is 3.
Len(FN)=2.  FN=[2,9].  FN[2]=9.  Len(SN)=3.  SN=-[5,7,0].  SN[3]=-0.
C[3]=0.  Since 9-0+0=9, 9<10, 9%10=9.  Len(A)=1.  A=[9].  Since
(9-9)/10=0, C[2]=0.
Len(FN)=1.  FN=[2].  FN[1]=2.  Len(SN)=2.  SN=-[5,7].  SN[2]=-7.
C[2]=0.  Since 2-7+0=-5, -5<-10, -5%-10=-5.  Len(A)=2.  A=[-5,9].  Since
(-5--5)/10=0, C[1]=0.
Len(FN)=0.  FN=[].  FN[0]=empty.  Len(SN)=1.  SN=-[5].  SN[1]=-5.
C[1]=0.  Since 0-5+0=-5, -5<-10, -5%-10=-5.  Len(A)=3.  A=[-5,-5,9].
Since (-5--5)/10=0, C[0]=0.
Len(FN)=0.  FN=[].  FN[0]=empty.  Len(SN)=0.  SN=-[].  SN[0]=empty.
Since both FN and SN are empty, next.  Since C[0]=0, the steps are done.
Since there are - in A, we check the sign of the last step A[1]=-5.
Since -5 is neg, we change the sign and process A from right to left.
A=[-5,-5,9]=-[+5,+5,-9].  C[3]=0.
Len(A)=3.  A=-[+5,+5,-9].  A[3]=-9.  Since -9<0, B=10, C[2]=-1.  Since
C[3]=0, thus -9+10+0=1.  Len(ANEW)=1.  ANEW=-[1].  C[2]=-1.
Len(A)=2.  A=-[+5,+5].  A[2]=+5.  Since +5>0, B=0, C[1]=0.  Since
C[2]=-1, thus 5+0-1=4.  Len(ANEW)=2.  ANEW=-[4,1].  C[1]=0.
Len(A)=1.  A=-[+5].  A[1]=+5.  Since +5>0, B=0, C[0]=0.  Since C[1]=0,
thus 5+0+0=5.  Len(ANEW)=3.  ANEW=-[5,4,1].  C[0]=0.
Len(A)=0.  A=-[].  Since A is empty, the problem is complete.  The final
Answer is -[5,4,1].
```

### B.3.2 CHAIN-OF-THOUGHT PROMPT FOR ADDITION-SUBTRACTION

**Problem: 128+367=?**
```
Explanation:  Let's think step by step.
128+367=128+300+67=428+67=495.  The final Answer is 495.
```
**Problem: 9980+29=?**
```
Explanation:  Let's think step by step.
9980+29=9980+20+9=10000+9=10009.  The final Answer is 10009.
```
**Problem: 29-570=?**
```
Explanation:  Let's think step by step.
29-570=29-500-70=-471-70=-541.  The final Answer is -541.
```
**Problem: -99-21=?**
```
Explanation:  Let's think step by step.
-99-21=-99-20-1=-119-1=-120.  The final Answer is -120.
```
**Problem: 483-389=?**
```
Explanation:  Let's think step by step.
483-389=483-300-80-9=183-80-9=103-9=94.  The final Answer is 94.
```
**Problem: -30+8002=?**
```
Explanation:  Let's think step by step.
-30+8002=-30+8000+2=-30+8002=7972.  The final Answer is 7972.
```

## B.4 MEMORIZED MULTIPLICATION PROMPT STRATEGIES

### B.4.1 CHAIN-OF-THOUGHT PROMPT FOR MEMORIZED MULTIPLICATION

For multiplication, we use prompt examples $128 * 367$ and $2035 * 87$ in order.

```
Q: 128*367=?
A: Let's think step by step.
128*367=128*(300+60+7)
128*367=128*300+128*60+128*7
128*367=38400+7680+896
128*367=46976
So, 128*367=46976.  The answer is 46976.
Q: 2035*87=?
A: Let's think step by step.
2035*87=2000*87+30*87+5*87
2035*87=174000+2610+435
2035*87=177045
So, 2035*87=177045.  The answer is 177045.
```

### B.4.2 ALGORITHMIC PROMPT FOR MEMORIZED MULTIPLICATION

```
Q: 128*367=
Explanation:
FN=128, FN=[1,2,8].  SN=367, SN=[3,6,7].  Len(FN)=3, Len(SN)=3.  Max len
is 3.  Since 3=3, the lengths of two numbers are equal, we pick FN and
break [1,2,8] into 3//3=1 group of three and one group of 3%3=0 leftover
digits.  Since there are 0 leftover digits, from [1,2,8] we break the
first 0 digits as [][1,2,8], thus the leftover group is []=empty and the
main group is [1,2,8].  Since there is 3//3=1 group of three, we break
the main group into 1 group of 3 each:  [1,2,8].  Reformatting for each
main group, we have 128.  Thus, ignoring the empty group, the groups are
128.  The other number is the MULVAL, thus MULVAL=367.
The submulproblems are 128*367=MUL1.  There is 1 mul operator.
**START**
Submulproblem:  128*367=MUL1
FN=128, FN=[1,2,8].  Mulval=367.  Len(FN)=3.  P0=0.
Len(FN)=3.  FN=[1,2,8].  FN[3]=8.  8*367=2936.  P0=0, append 0 zero [] to
[2,9,3,6][]:  [2,9,3,6]=ADV1.
Len(FN)=2.  FN=[1,2].  FN[2]=2.  2*367=734.  P0=1, append 1 zero [0] to
[7,3,4|0]:  [7,3,4,0]=ADV2.
Len(FN)=1.  FN=[1].  FN[1]=1.  1*367=367.  P0=2, append 2 zero [0,0] to
[3,6,7|0,0]:  [3,6,7,0,0]=ADV3.
Len(FN)=0.  Done.
++START++
Addition Problem:  ADV1+ADV2+ADV3=
Explanation:
The subproblems are ADV1+ADV2=ANS1, ANS1+ADV3=ANS2.  There are 2 add
operators.
Subproblem:  ADV1+ADV2=ANS1
FN=ADV1, FN=[2,9,3,6].  SN=ADV2, SN=[7,3,4,0].  Len(FN)=4, Len(SN)=4, max
len is 4.
Len(FN)=4.  FN=[2,9,3,6].  Len(SN)=4.  SN=[7,3,4,0].  FN[4]=6.  SN[4]=0.
C[4]=0.  6+0+0=6, 6<10, 6%10=6.  Len(A)=1.  A=[6].  (6-6)/10=0, C[3]=0.
Len(FN)=3.  FN=[2,9,3].  Len(SN)=3.  SN=[7,3,4].  FN[3]=3.  SN[3]=4.
C[3]=0.  3+4+0=7, 7<10, 7%10=7.  Len(A)=2.  A=[7,6].  (7-7)/10=0, C[2]=0.
Len(FN)=2.  FN=[2,9].  Len(SN)=2.  SN=[7,3].  FN[2]=9.  SN[2]=3.  C[2]=0.
9+3+0=12, 12>10, 12%10=2.  Len(A)=3.  A=[2,7,6].  (12-2)/10=1, C[1]=1.
Len(FN)=1.  FN=[2].  Len(SN)=1.  SN=[7].  FN[1]=2.  SN[1]=7.  C[1]=1.
2+7+1=10, 10=10, 10%10=0.  Len(A)=4.  A=[0,2,7,6].  (10-0)/10=1, C[0]=1.
Len(FN)=0.  FN=[].  Len(SN)=0.  SN=[].  Both are empty.  C[0]=1.  Not
done.  Len(A)=5.  ANS1=[1,0,2,7,6].  Since there are 2 add operators and
we processed up to ANS1, continue.  The new FN is [1,0,2,7,6].
###continued on next page
```

```
Subproblem: ANS1+ADV3=ANS2
FN=ANS1, FN=[1,0,2,7,6]. SN=ADV3, SN=[3,6,7,0,0]. Len(FN)=5, Len(SN)=5,
max len is 5. Len(FN)=5. FN=[1,0,2,7,6]. Len(SN)=5. SN=[3,6,7,0,0].
FN[5]=6. SN[5]=0. C[5]=0. 6+0+0=6, 6<10, 6%10=6. Len(A)=1. A=[6].
(6-6)/10=0, C[4]=0.
Len(FN)=4. FN=[1,0,2,7]. Len(SN)=4. SN=[3,6,7,0]. FN[4]=7. SN[4]=0.
C[4]=0. 7+0+0=7, 7<10, 7%10=7. Len(A)=2. A=[7,6]. (7-7)/10=0, C[3]=0.
Len(FN)=3. FN=[1,0,2]. Len(SN)=3. SN=[3,6,7]. FN[3]=2. SN[3]=7.
C[3]=0. 2+7+0=9, 9<10, 9%10=9. Len(A)=3. A=[9,7,6]. (9-9)/10=0,
C[2]=0.
Len(FN)=2. FN=[1,0]. Len(SN)=2. SN=[3,6]. FN[2]=0. SN[2]=6. C[2]=0.
0+6+0=6, 6<10, 6%10=6. Len(A)=4. A=[6,9,7,6]. (6-6)/10=0, C[1]=0.
Len(FN)=1. FN=[1]. Len(SN)=1. SN=[3]. FN[1]=1. SN[1]=3. C[1]=0.
1+3+0=4, 4<10, 4%10=4. Len(A)=5. A=[4,6,9,7,6]. (4-4)/10=0, C[0]=0.
Len(FN)=0. FN=[]. Len(SN)=0. SN=[]. Both are empty. C[0]=0. Done.
ANS2=[4,6,9,7,6]. Since there are add 2 operators and we processed up to
ANS2, complete. The final ADDAnswer is [4,6,9,7,6].
++END++
**END**
MUL1=[4,6,9,7,6]. Since there is 1 mul operator and we processed up to
MUL1, complete. We now combine the MUL results. Since 1 mul operator,
we append 3*(1-1)=3*0=0 zeros to MUL1, MUL1=[4,6,9,7,6][]=[4,6,9,7,6].
Addition Mul Problem: MUL1+EMPTY= Explanation: The subproblems are
MUL1+EMPTY=ANS1. There is 1 MA operator. Since EMPTY is in the
equation, ANS1=MUL1=[4,6,9,7,6]. Since there is 1 MA operator and we
processed up to ANS1, complete. The END Answer is [4,6,9,7,6].
```

## B.5 ALGORITHMIC PROMPT FOR PARITY

For parity, we use prompt examples [1, 1, 0, 1, 0] and [0, 1, 1, 0, 0, 0, 0, 0] in order.

```
Q: What is the parity on the list a=[1, 1, 0, 1, 0]?
A: We initialize s=
a=[1, 1, 0, 1, 0]. The first element of a is 1 so b=1. s = s + b = 0 +
1 = 1. s=1.
a=[1, 0, 1, 0]. The first element of a is 1 so b=1. s = s + b = 1 + 1 =
0. s=0.
a=[0, 1, 0]. The first element of a is 0 so b=0. s = s + b = 0 + 0 = 0.
s=0.
a=[1, 0]. The first element of a is 1 so b=1. s = s + b = 0 + 1 = 1.
s=1.
a=[0]. The first element of a is 0 so b=0. s = s + b = 1 + 0 = 1. s=1.
a=[] is empty. Since the list a is empty and we have s=1, the parity is
1.
```

## B.6 SCRATCHPAD PARITY FOR PARITY (ANIL ET AL., 2022)

```
Q: What is the parity on the list a=[1, 1, 0, 1, 0]?
A: [1, 0, 0, 1, 1], the parity is 1.
Q: What is the parity on the list a=[0, 1, 1, 0, 0, 0, 0, 0]?
A: [0, 1, 0, 0, 0, 0, 0, 0] , the parity is 0.
```

## B.7 ALGORITHMIC PROMPT FOR MULTI-ADD AND MULTIPLY-AS-ADD

**Problem: 128+367=**
Explanation:
The first number is 128, FN=[1,2,8]. The second number is 367,
SN=[3,6,7]. Since FN [1,2,8] has 3 digits, SN [3,6,7] has 3 digits, thus
the maximum number of digits is 3. In each subsequent step, we remove
one number from the end of FN and one from the end of SN.
Length of FN is 3. FN=[1,2,8]. Length of SN is 3. SN=[3,6,7].
FN[3]=8. SN[3]=7. C[3]=0. Since 8+7+0=15, 15>10, 15%10=5. Length of A
is 1. Thus A=[5]. Since (15-5)/10=1, C[2]=1.
Length of FN is 2. FN=[1,2]. Length of SN is 2. SN=[3,6]. FN[2]=2.
SN[2]=6. C[2]=1. Since 2+6+1=9, 9<10, 9%10=9. Length of A is 2. Thus
A=[9,5]. Since (9-9)/10=0, C[1]=0.
Length of FN is 1. FN=[1]. Length of SN is 1. SN=[3]. FN[1]=1.
SN[1]=3. C[1]=0. Since 1+3+0=4, 4<10, 4%10=4. Length of A is 3. Thus
A=[4,9,5]. Since (4-4)/10=0, C[0]=0.
There are no more digits and C[0]=0. Thus the process is complete. The
final Answer is [4,9,5].
**Problem: Problem: 9980+29=**
Explanation:
The first number is 9980, FN=[9,9,8,0]. The second number is 29,
SN=[2,9]. Since FN [9,9,8,0] has 4 digits, SN [2,9] has 2 digits, thus
the maximum number of digits is 4. In each subsequent step, we remove
one number from the end of FN and one from the end of SN.
Length of FN is 4. FN=[9,9,8,0]. Length of SN is 2. SN=[2,9].
FN[4]=0. SN[4]=9. C[4]=0. Since 0+9+0=9, 9<10, 9%10=9. Length of A
is 1. Thus A=[9]. Since (9-9)/10=0, C[3]=0.
Length of FN is 3. FN=[9,9,8]. Length of SN is 1. SN=[2]. FN[3]=8.
SN[3]=2. C[3]=0. Since 8+2+0=10, 10=10, 10%10=0. Length of A is 2.
Thus A=[0,9]. Since (10-0)/10=1, C[2]=1.
Length of FN is 2. FN=[9,9]. Length of SN is 0. SN=[]. FN[2]=9. SN
is empty. C[2]=1. Since 9+0+1=10, 10=10, 10%10=0. Length of A is 3.
Thus A=[0,0,9]. Since (10-0)/10=1, C[1]=1.
Length of FN is 1. FN=[9]. Length of SN is 0. SN=[]. FN[1]=9. SN is
empty. C[1]=1. Since 9+0+1=10, 10=10, 10%10=0. Length of A is 4. Thus
A=[0,0,0,9]. Since (10-0)/10=1, C[0]=1.
There are no more digits, but C[0]=1. Length of A is 5. Thus
A=[1,0,0,0,9]. The final Answer is [1,0,0,0,9].
**Problem: 802+7145+6=**
Explanation:
The subproblems are 802+7145=ANS1 and ANS1+6=ANS2. There are 2
operators.
Subproblem: 802+7145=ANS1
The first number is 802, FN=[8,0,2]. The second number is 7145,
SN=[7,1,4,5]. Since FN=[8,0,2] has 3 digits, SN=[7,1,4,5] has 4 digits,
thus the maximum number of digits is 4. In each subsequent step, we
remove one number from the end of FN and one from the end of SN.
Length of FN is 3. FN=[8,0,2]. Length of SN is 4. SN=[7,1,4,5].
FN[4]=2. SN[4]=5. C[4]=0. Since 2+5+0=7, 7<10, 7%10=7. Length of A
is 1. Thus A=[7]. Since (7-7)/10=0, C[3]=0.
Length of FN is 2. FN=[8,0]. Length of SN is 3. SN=[7,1,4]. FN[3]=0.
SN[3]=4. C[3]=0. Since 0+4+0=4, 4<10, 4%10=4. Length of A is 2. Thus
A=[4,7]. Since (4-4)/10=0, C[2]=0.
Length of FN is 1. FN=[8]. Length of SN is 2. SN=[7,1]. FN[2]=8.
SN[2]=1. C[2]=0. Since 8+1+0=9, 9<10, 9%10=9. Length of A is 3. Thus
A=[9,4,7]. Since (9-9)/10=0, C[1]=0.
Length of FN is 0. FN=[]. Length of SN is 1. SN=[7]. FN is empty.
SN[1]=7. C[1]=0. Since 0+7+0=7, 7<10, 7%10=7. Length of A is 4. Thus
A=[7,9,4,7]. Since (7-7)/10=0, C[0]=0.
There are no more digits and C[0]=0. Thus the process is complete.
Since there are 2 operators and we processed up to ANS1, there are more
operators to process. Thus, ANS1 is [7,9,4,7].
###continued on next page

```
Subproblem: ANS1+6=ANS2
The first number is ANS1, FN=[7,9,4,7]. The second number is 6, SN=[6].
Since FN=[7,9,4,7] has 4 digits, SN=[6] has 1 digit, thus the maximum
number of digits is 4.  In each subsequent step, we remove one number
from the end of FN and one from the end of SN.
Length of FN is 4.  FN=[7,9,4,7].  Length of SN is 1.  SN=[6].  FN[4]=7.
SN[4]=6.  C[4]=0.  Since 7+6+0=13, 13>10, 13%10=3.  Length of A is 1.
Thus A=[3].  Since (13-3)/10=1, C[3]=1.
Length of FN is 3.  FN=[7,9,4].  Length of SN is 0.  SN=[].  FN[3]=4.  SN
is empty.  C[3]=1.  Since 4+0+1=5, 5<10, 5%10=5.  Length of A is 2.  Thus
A=[5,3].  Since (5-5)/10=0, C[2]=0.
Length of FN is 2.  FN=[7,9].  Length of SN is 0.  SN=[].  FN[2]=9.  SN
is empty.  C[2]=0.  Since 9+0+0=9, 9<10, 9%10=9.  Length of A is 3.  Thus
A=[9,5,3].  Since (9-9)/10=0, C[1]=0.
Length of FN is 1.  FN=[7].  Length of SN is 0.  SN=[].  FN[1]=7.  SN is
empty.  C[1]=0.  Since 7+0+0=7, 7<10, 7%10=7.  Length of A is 4.  Thus
A=[7,9,5,3].  Since (7-7)/10=0, C[0]=0.
There are no more digits and C[0]=0.  Thus the process is complete.
Since there are 2 operators and we processed up to ANS2, the problem is
complete.  The final Answer is [7,9,5,3].
```
**Problem: 3*7=**
```
Explanation:
The subproblems are 3*7=MS1.  There is 1 * operator.
Subproblem:  3*7=MS1
Since the problem is multiplication, we find the smaller of the two
numbers and add the larger number as many times as the smaller number.
The first number is 3, FN=[3]=3.  The second number is 7, SN=[7]=7.
Since 3 is smaller than 7, we rewrite the problem as 7 summed together
3 times:  7+7+7.  We end at ANS(3-1)=2=ANS2.
The subproblems are 7+7=ANS1 and ANS1+7=ANS2.  There are 2 operators.
Subproblem:  7+7=ANS1
The first number is 7, FN=[7].  The second number is 7, SN=[7].  Since
FN=[7] has 1 digit, SN=[7] has 1 digit, thus the maximum number of digits
is 1.  In each subsequent step, we remove one number from the end of FN
and one from the end of SN.
Length of FN is 1.  FN=[7].  Length of SN is 1.  SN=[7].  FN[1]=7.
SN[1]=7.  C[1]=0.  Since 7+7+0=14, 14>10, 14%10=4.  Length of A is 1.
Thus A=[4].  Since (14-4)/10=1, C[0]=1.
There are no more digits and C[0]=1.  Length of A is 2.  Thus A=[1,4].
There are no more digits and the process is complete.  Since there are
2 operators and we processed up to ANS1, there are more operators to
process.  Thus, ANS1 is [1,4].
Subproblem:  ANS1+7=ANS2
The first number is ANS1, FN=[1,4].  The second number is 7, SN=[7].
Since FN=[1,4] has 2 digits, SN=[7] has 1 digit, thus the maximum number
of digits is 2.  In each subsequent step, we remove one number from the
end of FN and one from the end of SN.
Length of FN is 2.  FN=[1,4].  Length of SN is 1.  SN=[7].  FN[2]=4.
SN[2]=7.  C[2]=0.  Since 4+7+0=11, 11>10, 11%10=1.  Length of A is 1.
Thus A=[1].  Since (11-1)/10=1, C[1]=1.
Length of FN is 1.  FN=[1].  Length of SN is 0.  SN=[].  FN[1]=1.  SN is
empty.  C[1]=1.  Since 1+0+1=2, 2<10, 2%10=2.  Length of A is 2.  Thus
A=[2,1].  Since (2-2)/10=0, C[0]=0.
There are no more digits and C[0]=0.  Thus the process is complete.
Since there are 2 operators and we processed up to ANS2, the problem is
complete.  Since there is 1 * operator and we processed up to MS1, the
overall problem is complete.  The final Answer is [2,1].
```

## B.8 Chain-of-thought prompt for multi-add

We use the same prompt examples as the algorithmic prompt, which are $128 + 367$, $9980 + 29$, $802 + 7145 + 6$, $7 + 7 + 7$ in order.

```
Q: 802+7145+6=
A: Let's think step by step.
802+7145=7947
7947+6=7953
So, 802+7145+6=7953.  The answer is 7953.
Q: 7+7+7=
A: Let's think step by step.
7+7=14
14+7=21
So, 7+7+7=21.  The answer is 21.
```

### B.9   CHAIN-OF-THOUGHT PROMPT FOR MULTIPLY-AS-ADD

We use prompt examples $3 \times 107, 5 \times 6, 9 \times 9, 277 \times 2$ in order.

```
Q: 3*107=
A: Let's think step by step.
3*100=300
3*7=21
300+21=321
So, 3*107=321.  The answer is 321.
Q: 5*6=
A: Let's think step by step.
5*6=30
So, 5*6=30.  The answer is 30.
```

### B.10   ALGORITHMIC PROMPT FOR MULTI-ADD WITH ALGO CALLS

This prompt uses a single example to illustrate multi-number addition.  The special tokens that correspond to the start and end of the question extraction are `Subproblem:` and `<GET>`.

```
Problem: 802+7145+6=
Explanation:
The subproblems are 802+7145=ANS1 and ANS1+6=ANS2.  Since we ended on
ANS₂, there are 2 operators.
Subproblem:  802+7145⟨GET⟩=7947.  Since there are 2 operators and we
processed up to ANS1, there are more operators to process.
Subproblem:  7947+6⟨GET⟩=7953.  Since there are 2 operators and we
processed up to ANS2, the problem is complete.  The final Answer is
7953.
```

### B.11   ALGORITHMIC PROMPT FOR MULTIPLICATION-AS-ADDITION WITH ALGO CALLS

This prompt uses a single example to illustrate multiplication-as-addition, and combines it with the multi-number addition example from Section B.10.  The special tokens that correspond to the start and end of the question extraction are `Subproblem:` and `<GET>`.

```
Problem: 3*7=
Explanation:
Since the problem is multiplication, we find the smaller of the two
numbers and add the larger number as many times as the smaller number.
The first number is 3, FN=[3]=3.  The second number is 7, SN=[7]=7.
Since 3 is smaller than 7, we rewrite the problem as 7 summed together
3 times:  7+7+7.  We end at ANS(3-1)=2=ANS2.
The subproblems are 7+7=ANS1 and ANS1+7=ANS2.  Since we ended on ANS₂,
there are 2 operators.
Subproblem:  7+7⟨GET⟩=14.  Since there are 2 operators and we processed up
to ANS1, there are more operators to process.
Subproblem:  14+7⟨GET⟩=21.  Since there are 2 operators and we processed
up to ANS2, the problem is complete.  The final Answer is 21.
```

## B.12 CHAIN-OF-THOUGHT PROMPT FOR MULTI-ADD

```
Q: 9980+29=
A: Let's think step by step.
9980+29=10009
So, 9980+29=10009.  The answer is 10009.
Q: 802+7145+6=
A: Let's think step by step.
802+7145=7947
7947+6=7953
```

## B.13 ALGORITHMIC PROMPT FOR GSM8K

The following is the full prompt corresponding to the "W/ plan W/ algo" experiment in Figure 7(a).

```
Q: Tommy has 3 toy cars.  His neighbor, Jessie, has 3 cars too.  Jessie's
older brother has 5 more cars than Tommy and Jessie.  How many cars do
the three of them have altogether?
A: <NONALGO> Tommy and Jessie have 3+3=6 cars.  Jessie's brother has
5+6=11 cars.  Altogether, they have 6+11=17 cars.  The answer is 17.
Q: An electronic shop offers smartphones for $467 each, PCs are $128 more
expensive than smartphones, and advanced tablets are the prices of a
smartphone and a PC combined.  How much do you have to pay to buy one
of each of the three mentioned products?
A: <ALGO> To solve this problem, we need to find the prices of a PC and
an advanced tablet.  Then, we need to add the price of all three products
together.
The price of a PC is $128 more than a smartphone, thus the price of PC is
467+128.  We use the addition algorithm:
Problem:  467+128=
Explanation:
The subproblems are 467+128=ANS1.  There is 1 connecting operator.
Subproblem:  467+128=ANS1
The first number is 467, FN=[4,6,7].  The second number is 128,
SN=[1,2,8].  Since FN [3,6,7] has 3 digits, SN [1,2,8] has 3 digits, thus
the maximum number of digits is 3.  In each subsequent step, we remove
one number from the end of FN and one from the end of SN. Length of A is
0.
Length of FN is 3.  FN=[4,6,7].  FN[3]=7.  Length of SN is 3.
SN=[1,2,8].  SN[3]=8.  C[3]=0.  Since 7+8+0=15, 15>10, 15%10=5.  Length
of A is 1.  Thus A=[5].  Since (15-5)/10=1, C[2]=1.
Length of FN is 2.  FN=[4,6].  FN[2]=6.  Length of SN is 2.  SN=[1,2].
SN[2]=2.  C[2]=1.  Since 6+2+1=9, 9<10, 9%10=9.  Length of A is 2.  Thus
A=[9,5].  Since (9-9)/10=0, C[1]=0.
Length of FN is 1.  FN=[4].  FN[1]=4.  Length of SN is 1.  SN=[1].
SN[1]=1.  C[1]=0.  Since 4+1+0=5, 5<10, 5%10=5.  Length of A is 3.  Thus
A=[5,9,5].  Since (5-5)/10=0, C[0]=0.
There are no more digits and C[0]=0.  Thus the process is complete.
Since there is 1 operator and we processed up to ANS1, the problem is
complete.  The final Answer is [5,9,5].  Removing all 2 commas, we have
595.
The addition algorithm tells us that the price of a PC is 595.  Since
the price of an advanced tablet is the sum of a smartphone and a PC, its
price is 467+595.  We use the addition algorithm:  Problem:  467+595=
Explanation:
The subproblems are 467+595=ANS1.  There is 1 connecting operator.
Subproblem:  467+595=ANS1
The first number is 467, FN=[4,6,7].  The second number is 595,
SN=[5,9,5].  Since FN [4,6,7] has 3 digits, SN [5,9,5] has 3 digits, thus
the maximum number of digits is 3.  In each subsequent step, we remove
one number from the end of FN and one from the end of SN. Length of A is
0.
Length of FN is 3.  FN=[4,6,7].  FN[3]=7.  Length of SN is 3.
SN=[5,9,5].  SN[3]=5.  C[3]=0.  Since 7+5+0=12, 12>10, 12%10=2.  Length
of A is 1.  Thus A=[2].  Since (12-2)/10=1, C[2]=1.
Length of FN is 2.  FN=[4,6].  FN[2]=6.  Length of SN is 2.  SN=[5,9].
SN[2]=9.  C[2]=1.  Since 6+9+1=16, 16>10, 16%10=6.  Length of A is 2.
Thus A=[6,2].  Since (16-6)/10=1, C[1]=1.
Length of FN is 1.  FN=[4].  FN[1]=4.  Length of SN is 1.  SN=[5].
SN[1]=5.  C[1]=1.  Since 4+5+1=10, 10=10, 10%10=0.  Length of A is 3.
Thus A=[0,6,2].  Since (10-0)/10=1, C[0]=1.
There are no more digits, but C[0]=1.  Length of A is 4.  A=[1,0,6,2].
Thus the process is complete.  Since there is 1 operator and we processed
up to ANS1, the problem is complete.  The final Answer is [1,0,6,2].
Removing all 3 commas, we have 1062.
###continued on next page
```

```
 The addition algorithm tells us that the price of an advanced tablet
is 1062.  To buy one of each of these products, you would have to pay
467+595+1062.  We use the addition algorithm:
Problem:  467+595+1062=
Explanation:
The subproblems are 467+595=ANS1, ANS1+1062=ANS2.  There are 2 connecting
operators.
Subproblem:  467+595=ANS1
The first number is 467, FN=[4,6,7].  The second number is 595,
SN=[5,9,5].  Since FN [4,6,7] has 3 digits, SN [5,9,5] has 3 digits, thus
the maximum number of digits is 3.  In each subsequent step, we remove
one number from the end of FN and one from the end of SN. Length of A is
0.
Length of FN is 3.  FN=[4,6,7].  FN[3]=7.  Length of SN is 3.
SN=[5,9,5].  SN[3]=5.  C[3]=0.  Since 7+5+0=12, 12>10, 12%10=2.  Length
of A is 1.  Thus A=[2].  Since (12-2)/10=1, C[2]=1.
Length of FN is 2.  FN=[4,6].  FN[2]=6.  Length of SN is 2.  SN=[5,9].
SN[2]=9.  C[2]=1.  Since 6+9+1=16, 16>10, 16%10=6.  Length of A is 2.
Thus A=[6,2].  Since (16-6)/10=1, C[1]=1.
Length of FN is 1.  FN=[4].  FN[1]=4.  Length of SN is 1.  SN=[5].
SN[1]=5.  C[1]=1.  Since 4+5+1=10, 10=10, 10%10=0.  Length of A is 3.
Thus A=[0,6,2].  Since (10-0)/10=1, C[0]=1.
There are no more digits, but C[0]=1.  Length of A is 4.  A=[1,0,6,2].
Thus the process is complete.  Since there are 2 operators and we
processed up to ANS1, there are more operators to process.  The new FN
is [1,0,6,2].
Subproblem:  ANS1+1062=ANS2
The first number is ANS1, FN=[1,0,6,2].  The second number is 1062,
SN=[1,0,6,2].  Since FN [1,0,6,2] has 4 digits, SN [1,0,6,2] has 4
digits, thus the maximum number of digits is 4.  In each subsequent
step, we remove one number from the end of FN and one from the end of
SN. Length of A is 0.
Length of FN is 4.  FN=[1,0,6,2].  FN[4]=2.  Length of SN is 4.
SN=[1,0,6,2].  SN[4]=2.  C[4]=0.
Since 2+2+0=4, 4<10, 4%10=4.  Length of A is 1.  Thus A=[4].  Since
(4-4)/10=0, C[3]=0.
Length of FN is 3.  FN=[1,0,6].  FN[3]=6.  Length of SN is 3.
SN=[1,0,6].  SN[3]=6.  C[3]=0.  Since 6+6+0=12, 12>10, 12%10=2.  Length
of A is 2.  Thus A=[2,4].  Since (12-2)/10=1, C[2]=1.
Length of FN is 2.  FN=[1,0].  FN[2]=0.  Length of SN is 2.  SN=[1,0].
SN[2]=0.  C[2]=1.  Since 0+0+1=1, 1<10, 1%10=1.  Length of A is 3.  Thus
A=[1,2,4].  Since (1-1)/10=0, C[1]=0.
Length of FN is 1.  FN=[1].  FN[1]=1.  Length of SN is 1.  SN=[1].
SN[1]=1.  C[1]=0.  Since 1+1+0=2, 2<10, 2%10=2.  Length of A is 4.  Thus
A=[2,1,2,4].  Since (2-2)/10=0, C[0]=0.
There are no more digits and C[0]=0.  Thus the process is complete.
Since there are 2 operators and we processed up to ANS2, the problem is
complete.  The final Answer is [2,1,2,4].
Removing all 3 commas, we have 2124.
The addition algorithm tells us that the sum of all the products is 2124.
The answer is 2124.
Q: Cally and Danny washed their clothes.  Cally has 10 white shirts, 5
colored shirts, 7 pairs of shorts, and 6 pairs of pants, while Danny has
6 white shirts, 8 colored shirts, 10 shorts, and 6 pairs of pants.  How
many clothes did they wash?
A: <NONALGO> They washed 10+6=16 white shirts.  They washed 5+8=13
colored shirts.  They washed 7+10=17 shorts.  They washed 6+6=12 pants.
Therefore, Cally and Danny washed a total of 16+13+17+12=58 clothes.  The
answer is 58.
Q: If there are four times as many red crayons as blue crayons in a box,
and there are 3 blue crayons.  How many crayons total are in the box?
A: <NONALGO> There are 4 times as many red crayons as blue crayons, which
means there are 3+3+3+3=12 red crayons.  Since there are 3 blue crayons
and 12 red crayons, in total there are 12+3=15 crayons.  The answer is
15.
###continued on next page
```

```
 Q: Larry spent $5 for lunch and gave his brother $12.  How much did
Larry have at the beginning if he has $3 now?
A: <ALGO> To solve this problem, we need to find the total amount that
Larry spent and add it to the amount he has left.
Larry spent $5 for lunch and gave his brother $12, which means he spent
5+12.  We use the addition algorithm:
Problem:  5+12=
Explanation:
The subproblems are 5+12=ANS1.  There is 1 connecting operator.
Subproblem: 5+12=ANS1
The first number is 5, FN=[5].  The second number is 12, SN=[1,2].  Since
FN [5] has 1 digit, SN [1,2] has 2 digits, thus the maximum number of
digits is 2.  In each subsequent step, we remove one number from the end
of FN and one from the end of SN. Length of A is 0.
Length of FN is 1.  FN=[5].  FN[2]=5.  Length of SN is 2.  SN=[1,2].
SN[2]=2.  C[2]=0.  Since 5+2+0=7, 7<10, 7%10=7.  Length of A is 1.  Thus
A=[7].  Since (7-7)/10=0, C[1]=0.
Length of FN is 0.  FN=[].  FN[1]=0.  Length of SN is 1.  SN=[1].
SN[1]=1.  C[1]=0.  Since 0+1+0=1, 1<10, 1%10=1.  Length of A is 2.  Thus
A=[1,7].  Since (1-1)/10=0, C[0]=0.
There are no more digits and C[0]=0.  Thus the process is complete.
Since there is 1 operator and we processed up to ANS1, the problem is
complete.  The final Answer is [1,7].  Removing all 1 comma, we have 17.
The addition algorithm tells us that the amount spent is 17.  Larry has
$3 now, so he must have had 17+3 at the beginning.  We use the addition
algorithm:
Problem:  17+3=
Explanation:
The subproblems are 17+3=ANS1.  There is 1 connecting operator.
Subproblem: 17+3=ANS1
The first number is 17, FN=[1,7].  The second number is 3, SN=[3].  Since
FN [1,7] has 2 digits, SN [3] has 1 digit, thus the maximum number of
digits is 2.  In each subsequent step, we remove one number from the end
of FN and one from the end of SN. Length of A is 0.
Length of FN is 2.  FN=[1,7].  FN[2]=7.  Length of SN is 1.  SN=[3].
SN[2]=3.  C[2]=0.  Since 7+3+0=10, 10=10, 10%10=0.  Length of A is 1.
Thus A=[0].  Since (10-0)/10=1, C[1]=1.
Length of FN is 1.  FN=[1].  FN[1]=1.  Length of SN is 0.  SN=[].
SN[1]=0.  C[1]=1.  Since 1+0+1=2, 2<10, 2%10=2.  Length of A is 2.  Thus
A=[2,0].  Since (2-2)/10=0, C[0]=0.
There are no more digits and C[0]=0.  Thus the process is complete.
Since there is 1 operator and we processed up to ANS1, the problem is
complete.  The final Answer is [2,0].  Removing all 1 comma, we have 20.
The addition algorithm tells us that the total amount is 20.  The answer
is 20.
Q: Shawn has five toys.  For Christmas, he got two toys each from his mom
and dad.  How many toys does he have now?
A: <NONALGO> He has 5 toys.  He got 2 from mom, so after that he has
5+2=7 toys.  Then he got 2 more from dad, so in total he has 7+2=9 toys.
The answer is 9.
Q: Karen wanted to go out to get some fast food.  She pulls up to the
drive-through and orders a 5-dollar burger.  Her son then yelled out that
he wanted a 4-dollar sandwich, so it was added to the order.  Karen then
decided to order some drinks and opted for two 4-dollar smoothies.  What
is the total cost of Karen's fast-food order?
A: <NONALGO> Karen and her son order 5+4=9 dollars worth of food.  Karen
decides to buy 4+4=8 dollars worth of smoothies.  Thus, the total for
this order is 9+8=17 dollars.  The answer is 17.
Q: If there are 100 cars in the parking lot and 6 more cars arrive, how
many cars are in the parking lot?
A: <NONALGO> There are 100+6=106 cars in the parking lot.  The answer is
106.
```

