# OpenReview forum: "Teaching Algorithmic Reasoning via In-context Learning"
_ICLR.cc/2023/Conference — Submitted to ICLR 2023_

### Official Review · Reviewer_Ke6m · 2022-10-23

**Confidence:** 4
**Correctness:** 4
**Technical Novelty And Significance:** 2
**Empirical Novelty And Significance:** 2
**Recommendation:** 5

**Clarity, Quality, Novelty And Reproducibility:**

The proposed method is built upon Chain of Thoughts and scratchpad line of ideas but the specific method is novel to the best of my knowledge. The claims are supported by thorough experiments and results under each setting. And the paper is clearly written.

**Strength And Weaknesses:**

Strength:
- The method makes sense conceptually and demonstrates significant performance gain upon standard scratch pad/CoT in multiple aspects.
- The performance improvement for length extrapolations is especially impressive.
- The paper is clearly written with concrete examples.

Weaknesses:
- The demonstrated examples are all for very basic arithmetic operations, and it is unclear whether this technique can scale to more complicated cases.
- The algorithmic prompting method conceptually requires that we know how to break down the target operation (like parity) into recurrent steps and provide traces. With this level of specificity required, it is no longer necessary to use prompting to solve the tasks (we already solved it with our brain when providing the prompt and can totally write a program for it). Some generalizations (other than length) from the demonstration to the actual task would make the method much more useful. Some examples:
    - Can algorithmic prompting be extended to operations we only know the structure of? Like solving dynamic programming problems with demonstrations for a DP problem with different state space or update functions?
    - The skill composition could also be very useful if the demonstration of combining skills A and B can be generalized to combine A and C or even B and C.
- The length extrapolation intuitively only works because the recurrent step is the same for all lengths, so LM can follow the demonstration in a straightforward and local way. Although this is arguably possible for all operations, it could lead to a very large state to carry over in each recurrent step. In fact, even for just addition, the copying of a large number of digits in context repetitively already seems very expensive given the limited context window. The second-pass strategy seems a promising start to handle this. Maybe the demonstration can just only demonstrate the recurrent step from f(n + 1) to f(n), so that each recurrent step can be generated in a separate context like a recursive call?
- For the final tool use demonstration, what is the advantage of <call_add_algo> over calling the injected calculator as designed in the original dataset?

**Summary Of The Paper:**

This paper proposes a new prompting method called algorithmic prompting that aims to teach algorithms to LLMs via in-context learning. Specifically, the method focuses on using very details execution traces and explanations as part of the in-context demonstration to remove any ambiguity. Using this prompting method, in-context learning performance on simple arithmetic operations is significantly improved, especially under length extrapolation setting. Furthermore, the authors also demonstrate that models can acquire more/composed skills by having algorithmic prompting demonstrations of two different/composed algorithms in the context. The authors also show that it is possible to use the learned algorithm as a tool for solving math word problems.


**Summary Of The Review:**

Overall, the proposed method demonstrates significant improvement in some simple settings and I think this is an interesting direction. But so far it is unclear whether this technique can be scaled to more complicated problems.

---

> ### Author Response · Authors · 2022-11-15
> **Author Response Part 1**
>
> Thank you for your time and effort in reviewing our paper! We are happy to see your appreciation for the conceptual clarity and impressive performance of our method. If our response addresses your concern, we appreciate it if you increase your score accordingly. Otherwise, please let us know about your remaining concerns.
>
> >The demonstrated examples are all for very basic arithmetic operations, and it is unclear whether this technique can scale to more complicated cases or to solve dynamic programming problems.
>
> We agree with you that whether and how this method can scale to more complicated cases is an important research question. Nonetheless, we think that the current work still represents a substantial contribution. Although arithmetic and parity problems may seem “toy” for humans or computer programs, they have been long standing challenges for deep learning models. These tasks have been widely studied as marquee challenges in the field and their low performance has often been used as evidence for the lack of reasoning abilities in large language models (see e.g. [1]-[5]). Our work provides the **first** instance of general purpose language models being able to solve arithmetic problems robustly out of distribution, and provides proof that such reasoning abilities already exist in LLMs.
>
> >The algorithmic prompting method conceptually requires that we know how to break down the target operation (like parity) into recurrent steps and provide traces. With this level of specificity required, it is no longer necessary to use prompting to solve the tasks (we already solved it with our brain when providing the prompt and can totally write a program for it). Some generalizations (other than length) from the demonstration to the actual task would make the method much more useful.
>
> One thing we may want to ask the model to do is _algorithm discovery_. Indeed, we are not offering a solution to that in this work. Another thing we would want to see is if we give an algorithm to the model, that it can actually generalize by using this algorithm. We are the first to show that the model can apply algorithmic steps in a different context than what was seen in the prompt examples.
>
> Ultimately, we agree that solving questions where we can provide a general solution strategy to the model without detailing the exact implementation of the individual steps is an important goal of this research direction. We demonstrate a first example of this in the multiplication algorithm in Section 3.2, in which we give the model a high level strategy of breaking a problem down into smaller multiplication problems and recombining the results to get the final answer. In this experiment, we do not tell the model how to perform 1xn-digit multiplication, and instead rely on the model’s innate abilities to do so. Interestingly, we teach the model to break a large number of n digits into n/3 groups of 3 and 1 optional group with the remainder digits. For example 2035 would be broken into 2 and 035. However, there is only ever a single group of 3 in the prompt examples. Nonetheless, the model is able to generalize correctly and break long digits into multiple groups of 3 plus a remainder group. So the model could correctly break 28018399 into 280, 183, and 99. This provides another example of generalization that is not length generalization. Please see Section A.4 for more details.
>
> >The skill composition could also be very useful if the demonstration of combining skills A and B can be generalized to combine A and C or even B and C.
>
> Thank you for this insightful idea! To explore this direction, we created a new task of _multi-number subtraction_. In this case, we teach the model how to do addition-subtraction (in the combined prompt) as skill A, and multi-number addition as skill B, and see if it can perform well on multi-number subtraction as skill C. The model does not see any examples of multi-number subtraction in the prompt. We use the dialogue-approach to do this, and find that the model can indeed generalize to multi-number subtraction and outperform other baselines. Please see Figure 20 for the results.
>
> [1] https://arxiv.org/abs/2005.14165 Language Models are Few-shot Learners (Section 3.9.1)
>
> [2] https://arxiv.org/abs/2206.14858 Solving Quantitative Reasoning Problems with Language Models (Section 3.1)
>
> [3] https://arxiv.org/abs/2112.00114 Show Your Work: Scratchpads for Intermediate Computation with Language Models (Section 3)
>
> [4] https://arxiv.org/abs/2102.13019 Investigating the Limitations of Transformers with Simple Arithmetic Tasks
>
> [5] https://arxiv.org/abs/2202.07206 Impact of Pretraining Term Frequencies on Few-Shot Reasoning

---

> > ### Author Response · Authors · 2022-11-15
> > **Author Response Part 2**
> >
> > >The length extrapolation intuitively only works because the recurrent step is the same for all lengths, so LM can follow the demonstration in a straightforward and local way. Although this is arguably possible for all operations, it could lead to a very large state to carry over in each recurrent step. In fact, even for just addition, the copying of a large number of digits in context repetitively already seems very expensive given the limited context window. The second-pass strategy seems a promising start to handle this. Maybe the demonstration can just only demonstrate the recurrent step from f(n + 1) to f(n), so that each recurrent step can be generated in a separate context like a recursive call?
> >
> > Thanks for this suggestion! Performing each recurrent step in a separate context is a great idea, though it is not easily extensible from our current setup (as we have seen from second-pass, simply removing information from the context leads to degradations in performance). Nonetheless, we use a similar approach of separate context in the <call_add_algo> experiment. We have added new experiments that explore using separate contexts for composition tasks as a way to bypass context length limitations, and we find that this outperforms the second pass strategy and achieves near perfect performance on the composition tasks (see Figure 19).
> >
> > >For the final tool use demonstration, what is the advantage of <call_add_algo> over calling the injected calculator as designed in the original dataset?
> >
> > This is a good question. In the specific case of solving GSM8k arithmetic computations, there is no advantage. However, we are using GSM8k as a well-controlled test bed to evaluate this capability in LLMs (specifically, the capability of interfacing with copies of itself as a way to use separate contexts and different skill specializations).  The hope is that once we figure out how to equip LLMs with algorithmic reasoning abilities, we can use that in conjunction with the model’s abstract reasoning and general knowledge to go beyond what can be easily solved by an interpreter or calculator, such as simplifying math equations. This paper lays the starting point for that analysis.
> >
> > Thank you again for your review! We hope we were able to address your concerns and would welcome any further discussion if not. We kindly ask that you consider raising your score if your concerns were addressed.

---

### Official Review · Reviewer_9LM1 · 2022-10-23

**Confidence:** 4
**Correctness:** 4
**Technical Novelty And Significance:** 2
**Empirical Novelty And Significance:** 3
**Recommendation:** 5

**Clarity, Quality, Novelty And Reproducibility:**

Clarity can certainly be improved (see above). The technical quality and novelty is scarce, there is not actual learning but just interventions on the input prompt.

I would like to raise a concern about reproducibility of the experiments. I wanted to reproduce some of the experiments but found out that OpenAI’s Codex code-davinci-002 model is not publicly available yet, which is somewhat awkward. The best I could do was to try reproducing OOD results with the text-davinci-002 model, which exhibited similar reasoning processes but I got results (for 14 digits) that are much lower, i.e. 30%, than what is shown in Figure 3, i.e. approximately 97-98%. I tried with 10 samples where each addend has 14 digits, while providing the model with a single example of summation with 3-digit numbers.

Perhaps the authors can explain why this happened? Is that only due to a model difference or too few examples in the prompt? Please find here a txt file showing the prompt I used (the same as in section G.2) and the response I got from the text-davinci-002 model:
https://file.io/vqUWwuH8uZpq


**Strength And Weaknesses:**

The underlying idea of showing a language model how to break down “complex” data manipulation is quite interesting. From the results, the model seems to benefit from this prompting technique.

I found the paper somewhat hard to follow and the writing style rather confusing. For instance, I did not understand what “defining algorithms as skills” means exactly. This is mentioned several times in the manuscript but it looks to just boil down to using the aforementioned prompting technique. The paper should make a clear statement somewhere that the language model is not re-optimized and no weight updates actually happen, since the name “in-context learning” might be misleading.

The main drawback of the manuscript is that technical details of the experimentation are largely missing. The authors do not mention explicitly how many examples they provide in the prompt for each task, whether it is only one or multiple. Details of the baseline are not provided either, such as which prompts are used, making the comparison between different ideas harder.

Section 3, evaluation metric paragraph -- “We measure [..] OOD performance [..] where the model sees shorter examples at training time and generalize” -- missing s, typo -- ”to longer problems at test time”. This should be rephrased, there is no training/test time as long as a learning problem is not formulated and executed.

Usage of space is not well-balanced, the authors dedicated a full subsection for explaining “two-number addition” (section 3.1) and only a few paragraphs for all the others. In particular, the algorithm for parity is not even mentioned but its results appear in table 2.

Section 4 shows that we can “teach” multiple problems at a time, such as subtraction and addition. This looks to perform worse than showing each problem separately, therefore I am not exactly sure what conclusions can be drawn from there.

While it is undoubtedly interesting that “hard” reasoning capabilities emerge from LLMs with the right prompt, the problems faced in the paper seem only to be “toy problems”. The fact that this prompting fails on logical tasks (section 6) poses serious concerns on its applicability on harder problems other than simple mathematical operations and, therefore, its real-world utility.


**Summary Of The Paper:**

The authors study whether degrees of algorithmic reasoning emerge from pre-trained large language models (LLMs) by only intervening on the input prompt. Basically, the authors show that by describing all mathematical passages underlying a certain mathematical operation, e.g. addition, directly in the prompt, LLMs improve at solving these mathematical questions and they exhibit a problem solving process which is akin to an algorithm.

**Summary Of The Review:**

While I find the prompting scheme interesting, I don’t find the overall contribution to be relevant for ICLR. The technical contribution is poor, i.e. no learning whatsoever, and the manuscript is not precise and does not present experimental details very clearly. Lastly, a final concern about real-world utility as the prompting does not perform well on harder problems (logical tasks).

---

> ### Author Response · Authors · 2022-11-15
> **Author Response Part 1**
>
> Thank you for your time and effort in reviewing our paper! We appreciate your detailed review and your enthusiasm in reproducing our experiments. If our response addresses your concern, we appreciate it if you increase your score accordingly. Otherwise, please let us know about your remaining concerns/questions.
>
> >The paper should make a clear statement somewhere that the language model is not re-optimized and no weight updates actually happen, since the name “in-context learning” might be misleading.
>
> >The technical contribution is poor, i.e. no learning whatsoever
>
> >The technical quality and novelty is scarce, there is not actual learning but just interventions on the input prompt.
>
> The lack of weight updates seems to be one of the main concerns in this review. We want to point out that we had made the statement “in-context learning does not require any weight updates” in the first reference to in-context learning. To further improve clarity based on this feedback, we have emphasized this point in Figure 1 in the updated manuscript.
>
> We note that “in-context learning” (also known as prompting) is a very active research area and the term has been well-established with clear connotations of using the context information without re-optimizing model parameters. Prompting can be seen as an inference strategy for LLMs, which is a common topic of study at ICLR. E.g., a paper that studies in-context learning as Bayesian inference was recently published at ICLR 2022 [6]. Similar papers like [7] and [8] have also been recently published at NeurIPS. In-context learning has been explored by the field as an alternative to methods like fine-tuning (e.g. our performance on addition significantly outperforms those in the Scratchpad paper [3] which performs finetuning on addition traces), and it should be reasonable for these methods to be considered at the same venue. Thus, we think that the contribution of this paper should not be discounted on the basis of a lack of training / weight updates.
>
> >The main drawback of the manuscript is that technical details of the experimentation are largely missing.
>
> Thank you for pointing out the issue of missing experimental details. We have added more detail about the experimental setup throughout the paper and in dedicated sections (see Section A.2 for prompt statistics and Section B for prompt examples used for algorithmic prompting and other baselines for all the experiments in the paper). We hope the improved manuscript fully addresses this concern.
>
> **Regarding reproducibility:**
>
> We have added more details regarding the prompts to the updated paper in Appendix B. The OpenAI Codex model that we used (_code-davinci-002_) is in beta and is freely available upon request, which makes it one of the most accessible LLMs out there. The result you achieve is likely due to not using the full prompt (we use 3 examples for addition) and using the _text-davinci-002_ model (we did not benchmark on this model since it is not free to use). We have reproduced the addition prompt in full in Appendix B.1.1 and hope that this helps with reproducibility. Please let us know if you have further questions!
>
> >While it is undoubtedly interesting that “hard” reasoning capabilities emerge from LLMs with the right prompt, the problems faced in the paper seem only to be “toy problems”. The fact that this prompting fails on logical tasks (section 6) poses serious concerns on its applicability on harder problems other than simple mathematical operations and, therefore, its real-world utility.
>
> Although arithmetic and parity problems may seem “toy” for humans or computer programs, they have been long-standing challenges for deep learning models. These tasks have been widely studied as marquee challenges in the field (see e.g. [1]-[5]) and their low performance in OOD settings has often been used as evidence for the lack of reasoning abilities in LLMs.
>
> Our work provides the **first** instance of general purpose language models being able to achieve strong performance on OOD arithmetic problems, and provides proof that such reasoning abilities already exist in LLMs. This has significance in our understanding of the limitations and capabilities of current models, and points to ways of eliciting more capabilities from models in future work. Ultimately we would want to scale these approaches to more complex tasks, and we think that this work presents a promising start to a historically difficult challenge.
>
> Regarding the failure on logical tasks (GSM8k), we want to clarify that the failure isn’t on the logical reasoning aspect of GSM8k (which we use the chain-of-thought method for, _not_ our algorithmic prompting method), nor on the arithmetic reasoning part (which we use algorithmic reasoning for and achieve good performance on). Our result identifies and characterizes an interference phenomenon when different skills are combined within the same context, which sheds light on a _general_ limitation of LLMs.

---

> > ### Author Response · Authors · 2022-11-15
> > **Author Response Part 2**
> >
> > >Section 3, evaluation metric paragraph [...] This should be rephrased, there is no training/test time as long as a learning problem is not formulated and executed.
> >
> > Thank you for pointing this out! We have rewritten that section in the manuscript to clarify these concepts. We reproduce the section below:
> >
> > “We measure both in-distribution and OOD performance in all experiments.  For the in-context learning setting considered in this work, the data distribution is determined by the answer lengths of the prompting examples. Thus, questions with answer lengths that fall within those seen in the prompt are considered in-distribution, and those with longer lengths are considered out-of-distribution. The choice of length is natural given that it is a measure of complexity in the tasks we consider, and length generalization has a rich history as a measure of systematic generalization (Csordas et al., 2021; Anil et al., 2022). Thus, length generalization provides a good indication for whether the model has learned the underlying algorithm.”
> >
> > >Usage of space is not well-balanced, the authors dedicated a full subsection for explaining “two-number addition” (section 3.1) and only a few paragraphs for all the others. In particular, the algorithm for parity is not even mentioned but its results appear in table 2.
> >
> > We have now added clear reference to further ablations and the actual prompts used for each experiment, we hope that the additional detail helps to clarify these sections (e.g. see Section A.4 and Section B). Regarding the use of space, we chose to use the two-number addition task to explain the intuition behind our approach and include a number of ablation studies to understand the behavior of algorithmic prompting. The other sections follow the same intuition / strategy as those laid out in 3.1 and are meant to illustrate performance on a variety of tasks, so we opted to include their details in the appendix due to space constraints. We hope the newly added details make these sections more informative.
> >
> > >Section 4 shows that we can “teach” multiple problems at a time, such as subtraction and addition. This looks to perform worse than showing each problem separately, therefore I am not exactly sure what conclusions can be drawn from there.
> >
> > The goal of Section 4 is to understand the model’s behavior when multiple skills are taught simultaneously. This is relevant in cases when a task requires the use of different skills depending on the test question. We had added new analysis to better understand the Section 4 results (see Figure 14). Our findings suggest that there is minimal interference in this setting and even evidence of positive transfer (thus the performance is not worse than showing each separately), which we intuitively attribute to the similarity between the two algorithms. To further situate this understanding, compare this result to the interference issue identified in GSM8k. Intuitively, the core difference between these two settings is that the skills rely on very different capabilities (informal reasoning vs algorithmic execution). This provides us with an understanding of how and when interference happens, which we hope will be further studied and improved in future work.
> >
> > Thank you again for your review! We hope we were able to address your concerns and would welcome any further discussion if not. We kindly ask that you consider raising your score if your concerns were addressed.
> >
> >
> > [1] https://arxiv.org/abs/2005.14165 Language Models are Few-shot Learners (Section 3.9.1)
> >
> > [2] https://arxiv.org/abs/2206.14858 Solving Quantitative Reasoning Problems with Language Models (Section 3.1)
> >
> > [3] https://arxiv.org/abs/2112.00114 Show Your Work: Scratchpads for Intermediate Computation with Language Models (Section 3)
> >
> > [4] https://arxiv.org/abs/2102.13019 Investigating the Limitations of Transformers with Simple Arithmetic Tasks
> >
> > [5] https://arxiv.org/abs/2202.07206 Impact of Pretraining Term Frequencies on Few-Shot Reasoning
> >
> > [6] https://openreview.net/pdf?id=RdJVFCHjUMI An Explanation of In-context Learning as Implicit Bayesian Inference
> >
> > [7] https://arxiv.org/abs/2201.11903 Chain of Thought Prompting Elicits Reasoning in Large Language Models
> >
> > [8] https://arxiv.org/abs/2207.04901 Exploring Length Generalization in Large Language Models

---

> > > ### Comment · Reviewer_9LM1 · 2022-11-18
> > > **Thanks for the rebuttal**
> > >
> > > Many thanks for your articulated rebuttal which certainly helps in shedding light on some of my earlier concerns, namely:
> > > * reproducibility is now much much stronger
> > > * I appreciate the clarifications on Section 6: while reading the paper I was under the impression that you were solving logic tasks through prompting, but I know get it is sort of a secondary task testing the ability to do (again) arithmetic reasoning within a logic task. This is now clearer but also makes me think this is again another experiment limited to showing solely arithmetic reasoning.
> > > * ok I buy your argument on prompting being somehow present in learning conferences
> > >
> > > This said, I think this is a nice to read work but still too limited in scope. I understand you claim you are the first to perform arithmetic reasoning with prompting on language models, but this needs to be further discussed also in the light of the "public" comment just received. Aside from this, I am not completely sure the work shows true potential for being general: everything in the paper revolves around teaching simple arithmetic operations by prompting. How can this generalize? For instance, how do you expect to leverage such an approach to teach even the simplest of the sorting algorithms, not to mention an algorithm solving a combinatorial problem? I know one needs to start from simple tasks at the beginning but should also be able to show a potential for generalizing the approach, otherwise it gives the feeling of a dead-end paper.

---

> > > > ### Author Response · Authors · 2022-11-22
> > > > **Author response**
> > > >
> > > > Thank you for your response. From your original review and current response, it appears that we have addressed the main drawback originally raised by you and almost all of your concerns in the original review (except perhaps your concern about real world utility which you have reiterated in your response). Given this, we are somewhat confused that our response has not led to any change in your overall assessment/score. We respectfully ask that you consider evaluating our paper in that light.
> > > >
> > > > Below, we discuss your concerns in your recent comment:
> > > >
> > > > Regarding the public comment, please see our response to that comment. We are certainly not the first to _try_ to do addition with prompting, there has been a lot of interest in solving this problem but it has until now been unsuccessful. We provide a comparison of our method to the one suggested in the blogpost. We find that their method achieves <15% accuracy on length 7 while our performance is ~100%. At length 12 their performance drops to 0% while ours remain close to 100%. Our results are categorically better than the results in the blogpost, which reflects the difference in strategy and insights used in the respective works. See Figure 23 of the manuscript for details.
> > > >
> > > > We respectfully disagree with characterizing our paper as a “dead-end paper”. Our work is significant because this type of ability wasn’t thought to be possible in LLMs. We have shown that LLMs *can* perform symbolic, algorithmic manipulations that generalize far out of distribution. Moreover, one of the core aims of this work is to **understand** how LLMs behave under different types of prompting and we have a large number of experiments and ablations to that effect. One of the higher level insights from this work is that we can increase our control over the model’s behavior by increasing the specificity of the prompting information, which we believe will be very useful for prompting in general.
> > > >
> > > > We agree that there is still a long way to go towards general and robust reasoning ability in LLMs, but it is not a fair expectation to require a single paper to solve a whole area of research in one step. There are a number of parallel research directions that can influence and unlock further capabilities of algorithmic prompting. Efforts of scaling to much larger context lengths are underway, as seen by the already increasing context lengths of successive LLM releases. Memory-based mechanisms also simulate longer context, and it would be interesting to study how algorithmic prompting can be made to efficiently interface with these methods. Algorithmic prompting directly benefits from longer context or simulated context (e.g. memorizing transformers) and is thus a promising complement to these other lines of research. Given our insights on prompting, one may also look to study automatic ways of generating more robust and unambiguous explanations in other tasks and see how that affects performance. Various forms of context distillation may also be used to increase the complexity of tasks that we can solve.
> > > >
> > > > In this paper, we demonstrate the existence of a surprising and desirable capability in LLMs and present a concrete strategy to extract this behavior. We believe and hope you’ll agree that this is a valuable contribution and the community will find these results exciting and that this work enables the community to consider new research questions.

---

> > > > > ### Comment · Reviewer_9LM1 · 2022-11-24
> > > > > **Thanks for the response**
> > > > >
> > > > > I appreciated both the response to my comment and the point about the public comment. There is no need to be puzzled: I have already declared my intentions about changing the score in the private comments, but I had been awaiting the response to the public comment to do so.
> > > > >
> > > > > This said, I am not wishing to characterize your paper as a dead-end nor I had anywhere hinted at the fact that you need to solve a whole area. Rather, what I believe it would be important to have in the paper is a convincing discussion of how the results (which are quite nice) on a single task (difficult perhaps, but limited in complexity) can be generalized and used by the community to "unlock further capabilities of algorithmic prompting" in larger scale contexts. I see that you are placing much focus now on the insight you provide on how LLMs behave under different types of prompting, but the extent to which these insight exceed the specific task is not yet clear to me.

---

> > > > > > ### Author Response · Authors · 2022-11-28
> > > > > > **Response and proposed improvement to the manuscript**
> > > > > >
> > > > > > Thank you for your response and clarifications. We agree that a convincing discussion of how the results and insights can be generalized will help improve this paper. We propose to add the following section in the final manuscript, please let us know your feedback and if this helps address your concerns.
> > > > > >
> > > > > > >We have shown that LLMs are capable of executing arithmetic algorithms in a way that generalizes reliably out-of-distribution. The success of algorithmic prompting on arithmetic and parity tasks showcases the LLM’s potential to pick up logical patterns and use it to generalize at a mechanistic level. Our analyses uncover the following general insights about in-context learning: 1) Increasing the amount of detail and specificity helps to constrain the model’s interpretation of the prompting information, and therefore allow us to increase the reliability of the model’s outputs. Thus, one should specify unambiguous logical rules whenever possible when prompting. 2) When learning multiple algorithms that share similar subroutines together, there is synergy between the algorithms such that we can use fewer total prompt examples than would be required to learn each skill separately. 3) Learning skills that are very different in nature can lead to interference between the skills. We conjecture that this is due to the noise coming from soft attention and represents a fundamental limitation to current forms of in-context learning.
> > > > > >
> > > > > > >In this work, we evaluated algorithmic prompting on simple algorithms like parity, addition and multiplication. An important continuation of this work is to investigate how this approach can scale to more complex tasks of practical interest. There are many algorithmic tasks that are natural choices for future research, such as taking derivative or integral, simplifying equations, finding roots, and using induction as a proof technique. We believe that teaching such skills are possible given the positive results in this paper. Our results heighten the value of a number of research directions that can unlock further capabilities of algorithmic prompting. Firstly, the interference issue raises the importance of modularization for in-context learning. We proposed a dialogue-like approach where the model learns to use separate contexts when different skills are required, and showed that it has promising results in modularizing the skills and reducing the context length requirements. This type of composable communication between models loaded with different prompts opens up new avenues to explore. Moreover, the current approach is limited by context length. We have shown how context length can be directly leveraged to improve performance on algorithmic reasoning tasks, which means that an important research direction is increasing the effective context length of LLMs. Work such as memorizing transformers [1] is one step in this direction. On the other hand, finding ways of distilling an algorithmic prompt into fewer tokens will also unlock significant value. New forms of context distillation may also be able to encode the generalization behavior of the prompted model into a more permanent skill in the form of weights updates. Algorithmic prompting offers an existence proof for what algorithmic reasoning behavior can look like in LLMs, and we hope that being able to study this behavior will allow the community to find more efficient ways of extracting this capability.
> > > > > >
> > > > > >
> > > > > > [1] Memorizing Transformers. Yuhuai Wu, Markus N. Rabe, DeLesley Hutchins, Christian Szegedy. https://arxiv.org/abs/2203.08913

---

> > > > > > > ### Comment · Reviewer_9LM1 · 2022-11-28
> > > > > > > **About the proposed discussion**
> > > > > > >
> > > > > > > Dear Authors,
> > > > > > > many thanks for your response. This is certainly a much welcome addition to the work as it provides a neat summary of conclusions and hints that in principle should be possible to port to different tasks. I have also appreciated the attempt at identifying relevant tasks which can be used to extend the work.
> > > > > > > I will take this into consideration on the internal discussion.
> > > > > > > Regards

---

### Official Review · Reviewer_UYXX · 2022-10-25

**Confidence:** 3
**Correctness:** 3
**Technical Novelty And Significance:** 2
**Empirical Novelty And Significance:** 3
**Recommendation:** 8

**Clarity, Quality, Novelty And Reproducibility:**

The paper is well-written, the method is clear, and the analysis is well-organized. The work is as original as it can be given this is such a popular topic. The paper didn't mention if the code would be distributed, but I assume so. End of the day, it's a set of templates.

My main concern is presenting some blurry methods as contributions. I think the paper would be more honest by discussing harder the number of tokens and how the method cannot be generalized properly. I'm not convinced that the experiments in GSM8k add clear insights beyond addition. The solution of calling another model contradicts slightly a comment about "external tools" right before the subsection "Contributions" on page 2. Perhaps there is a missing lesson on how the limitation of the number of tokens can be alleviated by composition.

Additional comments:
- Given the workarounds for dealing with the max amount of tokens, I'd like to see a reflection on the potential limitations of LLM reasoning. There are models that accept more tokens, and some that claim to accept an unbound number (I expect them to deteriorate). Given how much space it took to increase the accuracy, we have to wonder if the max token is a solid ceiling for more precise reasoning.
- Page 6. "Length 8"  in the caption of table 2 probably refers to multiplication.

**Strength And Weaknesses:**

Strengths
- The systematic study of the structure of the prompt, including influential related work.
- Focus on the number of digits to study the generalization
- Study of kind of errors in the proposed prompt strength the proposal

Weaknesses
- The interference found is a negative result in an area poorly understood. It's entirely possible that a slight variation of the prompt managed to fix the issue. I'm not ready to accept strong interpretations of "X prompt didn't work". I suggest lowering the tone of such findings.
- The jump from addition to combination with subtraction is relatively small at an abstract level: just four combinations. This could become a strength if presented as a simple change that becomes challenging for the LM.

Potential weakness
- The study of the composition of skills gets confusing around multiplication. The setting of using a shorter explanation becomes out of the general hypothesis. This might become a strength if it were presented as a limitation and not as a solution.

**Summary Of The Paper:**

The paper introduced a more detailed prompt for LLM math reasoning, and study their impact on task composition and assisting in solving math problems in a recent dataset. The proposed method shows very high accuracy in simple tasks and starts to fall short when the tasks are more complex. One of the challenges is the max number of tokens that the model accepts, so they study multiple alternatives.
In summary, the paper shows that precision on the prompt can help to improve LLM reasoning, while the method requires longer prompts so it becomes harder to realize its gains.

**Summary Of The Review:**

The paper proposes a prompting method that focuses on providing a more precise text description of the reasoning. It performs very well in addition, parity and others, but space limitation of the context of LLM precludes performing the real experiments when doing composition and the MATHS problems.

I recommend the acceptance of the paper based on the positive findings, while I recommend interpreting the results without assuming that LLMs would be able to achieve high performance.

---

> ### Author Response · Authors · 2022-11-15
> **Author Response Part 1**
>
> Thank you for your time and effort in reviewing our paper! We are delighted by your vote of confidence in this work! We would like to address the comments and questions you’ve raised, and we’d be very open to further discussion.
>
> >The interference found is a negative result in an area poorly understood. It's entirely possible that a slight variation of the prompt managed to fix the issue. I'm not ready to accept strong interpretations of "X prompt didn't work". I suggest lowering the tone of such findings.
>
> This is a fair question. We did try a number of variations in the prompt to try to remove the interference issue, such as:
>
> - Including algorithmic output in all prompt examples
>
> - Changing the ordering of algo vs nonalgo examples
>
> - Using different phrasing for extracting the answer from the algo output for use in further informal reasoning
>
> - Providing clear delineation / separation between informal reasoning and algorithmic reasoning outputs
>
> - Instructing the model to ignore the algorithmic outputs when doing informal reasoning steps
>
> - Using indents to make the algorithmic output more code like
>
> - Rephrasing the algorithmic steps to be more code-like and less natural language-like
>
> - Rephrasing the algorithmic steps to be more natural language-like and less code-like
>
> What we presented in the paper was the best / simplest version after this investigation. With this, we hope you’d agree that the interference issue (at least for this task) cannot be easily resolved through simple prompt changes, and presents an interesting limitation of the existing models.
>
> >The jump from addition to combination with subtraction is relatively small at an abstract level: just four combinations. This could become a strength if presented as a simple change that becomes challenging for the LM.
>
> Thank you for this suggestion. Addition and subtraction are indeed similar to each other, but the two algorithms present real differences that the model has to learn. In particular, the subtraction algorithm (see Section B.3.1) involves checking the signs of each digit in the answer from the first pass and performing a secondary set of steps if any digit is negative. The model has to determine the right processing path based on the individual test question. It is unknown a priori whether this would work well, and our goal was to analyze the model’s behavior. Our results show that this _can_ be done for algorithms like addition and subtraction. Nonetheless, scaling to more algorithms or different types of algorithms (e.g. see interference issue in GSM8k) may require more efficient strategies. Indeed, the prompt development was indeed nontrivial, and we have added some more discussion related to this point in Section A.5.
>
> >My main concern is presenting some blurry methods as contributions. I think the paper would be more honest by discussing harder the number of tokens and how the method cannot be generalized properly.
>
> >Given the workarounds for dealing with the max amount of tokens, I'd like to see a reflection on the potential limitations of LLM reasoning. There are models that accept more tokens, and some that claim to accept an unbound number (I expect them to deteriorate). Given how much space it took to increase the accuracy, we have to wonder if the max token is a solid ceiling for more precise reasoning.
>
> Thank you for this suggestion. We’ve added a discussion around the implications of the context length limitation in the conclusion. Our work demonstrates an ability to directly improve algorithmic reasoning performance by increasing the amount of context used. The core of our contribution is in studying the new state of capabilities and limitations in current LLMs given the improvements unlocked by algorithmic prompting. Our results raise the importance of increasing context length and other ways of simulating long context as promising research directions. We note that context length isn’t the only way to allow the model to attend to long traces of information; it could also be tackled through things such as recurrence, external memory, or other context distillation techniques. Moreover, there are other factors contributing to performance, such as the pretraining strategy and model architectures.

---

> > ### Author Response · Authors · 2022-11-15
> > **Author Response Part 2**
> >
> >
> > >I'm not convinced that the experiments in GSM8k add clear insights beyond addition. The solution of calling another model contradicts slightly a comment about "external tools" right before the subsection "Contributions" on page 2. Perhaps there is a missing lesson on how the limitation of the number of tokens can be alleviated by composition.
> >
> > Thank you for raising this point. Indeed, the GSM8k experiments can be considered a form of tool use. We draw the distinction between this type of tool use (where models are interfacing with copies of itself as a way to use separate contexts) with what we call “external tool use” (where the model can access things like a calculator or an interpreter) to emphasize that we are studying addition as an simplified proxy for the type of reasoning that we would want the model to do eventually. An example of a task that can’t be easily solved by a python program is simplifying math equations. Addition is a good task for scientific study because it is easy to define and to evaluate, and yet is extremely challenging for LLMs. However, once we can equip LLMs with algorithmic reasoning abilities, the goal would be to use them in conjunction with the model’s soft reasoning and general knowledge to go beyond what can be easily solved by an interpreter. This paper lays the starting point for that direction.
> >
> > As you suggested, calling models with separate contexts is a potential way of alleviating the context length limitation and interference issue we’ve identified. We have added new results to demonstrate this effect concretely, which breaks the composition tasks down and uses separate contexts to solve the individual components (see Section A.6 and Figure 19). We present this method as an alternative to the second pass strategy for dealing with context length limitations.

---

> > > ### Comment · Reviewer_UYXX · 2022-11-22
> > > **Thank you**
> > >
> > > Thank you for your comments.
> > > It seems the paper is heading to rejection.
> > > I see some merit on how narrow is the class of problems, especially in comparison with the literature.
> > > (To be honest, this is all because we are using learning for the tasks. The tasks per see are so small).

---

> > > > ### Author Response · Authors · 2022-11-23
> > > > **Author response**
> > > >
> > > > Thank you for your response. In our view, the narrowness of the class of problems alone should not be grounds for rejection. There are a huge number of analysis papers that evaluate on toy/synthetic tasks to gain insights into how a model works. Our work is primarily about understanding and laying the groundwork for a different learning paradigm, which necessitates the use of well-controlled tasks to facilitate analysis. We have the additional benefit of also introducing a method that achieves significant performance boosts on a set of widely benchmarked tasks, but the fact that this method hasn’t been adapted to and tested on all other tasks yet should not distract from the main contributions of the paper.
> > > >
> > > > Learning to do arithmetic has been a long standing problem for language models and is often used to demonstrate their weakness in tasks that are considered simple for most people. Our work is the first to show that there is no fundamental limitation in language models that prevents them from learning these algorithms. In this work, we demonstrate that teaching algorithms in context is possible and we identify and study 4 key learning stages. This opens up possibilities for a drastically different view to learning. However, expecting us to go all the way to teaching super complex algorithms in one paper is unrealistic given that we know even addition was long considered a very challenging task for language models.

---

> > > > > ### Comment · Reviewer_UYXX · 2022-11-23
> > > > > **I understand**
> > > > >
> > > > > Yes, the narrowness of the class of problem should not be ground for rejection. One big issue is that the literature is hard to compare. Perhaps an improved version of the paper should emphasize even more the challenges of the task.

---

### Author Response · Authors · 2022-11-15
**Summary of new experiments and modifications**

Hi all! We want to summarize the key changes that were made in the rebuttal revision.

The following is a list of modifications we have made to the manuscript per reviewer feedback:

- Added details regarding the datasets and prompts used for the experiments throughout the paper and in dedicated sections. See Section A.2 and Section B.

- Added discussion around scaling to the accumulation of more algorithms in Section A.5.

- Added discussions around implications for future research directions in the conclusion.

- Added information regarding the extrapolation required by the multiplication algorithm in Sections A.4 and B.4.2.

- Clarified that in-context learning does not require weight updates in Figure 1 and in Introduction.

- Rewrote the evaluation metric section to clarify what we mean by out-of-distribution.

The following is a list of new experiments we have added to the manuscript per reviewer feedback:

- New analysis for skill accumulation section (Figure 14), which shows that there is minimal interference in this setting and even evidence of positive transfer.

- New experiments to show that calling on models loaded with different contexts is an effective way to move past context length limitations and achieve near perfect performance on the composition tasks (Figure 19).

- New experiments to show that combining skills A and B can be generalized to combining B and C in the multi-number composition setting (Figure 20).

---

### Public Comment · ~Ekin_Akyürek1 · 2022-11-17
**prompt design**

Dear authors,

I am really sorry to be that person that points out their own work. But, I see that some of the prompt designs proposed in this paper have similarities to what we've discussed in our analysis post. We had analyzed & improved scratchpads/CoTs for the addition task in the following post:

Title: Notes on Teaching GPT-3 Adding Numbers
Link: https://lingo.csail.mit.edu/blog/arithmetic_gpt3/

Please see the prompts and code that I used to produce these results:
Code: https://github.com/ekinakyurek/gpt3-arithmetic

For example, the authors discuss that the fundamental issue with scratchpad style prompts is that the rules can be ambigious to the model and the scratchpad should be annotated with natural language so that model can infer the rules better:

>  But from the scratchpad-style example the model could have concluded that the carry is 1 whenever we add two even digits together and 0 otherwise, or that the first digit-pair generates a carry of 1, the
second digit-pair generates a carry of 0, and so on. In order for the model to extrapolate the correct
pattern, it must be biased in such a way that the general and correct rule is the default interpretation.
Such alignment, however, can not be reliably expected from current mode

Something similar we say in our post:
>... a scratchpad must not leave anything to interpretation by the model and place appropriate markers. For instance, the tri-way sum without carry-over markers (1+8+9) requires the model to infer which summand is the carry-over, and thus results in poor performance.

Overall, our prompts in the post already adds natural language explanations to the scrachpad-prompts to improve the performance in the addition task --- this shared feature is what the authors listed first in the contribution section:

> We introduce Algorithmic Prompting, which involves providing a detailed description of
the algorithm execution on running examples, and **using explicit explanation and natural
language instruction to remove ambiguity**. For a comparison of algorithmic prompting to
existing prompting techniques, see Section 2 and Table 1.

I appreciate if you could've discussed this in the paper and compare given these similarities.


Best regards

---

> ### Author Response · Authors · 2022-11-22
> **Author response**
>
> Thank you for your comment. We’d like to point out a few key differences between our work and the analysis in the blog for the sake of the review process.
>
> 1. This blogpost evaluates variants of the scratchpad prompt on two-number addition. Our paper studies teaching algorithms such as addition, subtraction, multiplication, and parity as skills, and explores three additional settings: skill accumulation, skill composition, and skill as tool use. Thus, we are covering many different areas. However, even if we focus just on two-number addition, the details of our strategies are different which have resulted in categorically different performances between our works. At length 7, the performance of the blog prompt is ~15% while our prompt achieves ~100%. At length 12, the blog prompt is at ~0% accuracy while ours is still ~100%. We have included a detailed comparison in the paper (Figure 23).
>
> 2. The passage you quoted is taken out of context and does not accurately present our main contributions. The explicit explanation and patterns in our work refer to the rule definitions of each algorithmic step. We have shown in several ablation studies that these explicit rules are primarily responsible for the performance improvements. In Figure 3a, we showed that removing these rules leads to significant degradations in performance, even though we keep our detailed formatting and natural language instructions. In Figure 24, we showed that a prompt with inconsistent carry-over markers but more rule inclusions significantly outperforms the prompt with consistent carry-over markers.
>
> 3. Although we both discuss the benefit of using natural language descriptions, our work highlights their usefulness more for explaining abstract concepts that are harder to illustrate, such as the grouping operation in memorized multiplication (see Section 3.2 and A.4). For the simpler task of addition, we have shown in Figure 9 that a symbols-only addition prompt with minimal natural language still performs significantly better than all baselines and achieves length generalization, again showing the importance of explicit rule definitions.
>
> We have added a discussion and included references to the blog in the updated manuscript.

---

### Decision · Program_Chairs · 2023-01-20

**Decision:**

Reject

**Justification For Why Not Higher Score:**

Not sufficient support from the reviewers.

**Justification For Why Not Lower Score:**

NA

**Metareview: Summary, Strengths And Weaknesses:**

This paper proposes a set of prompts to enable the use of large language models (LLMs) to answer algorithmic reasoning questions (addition, multiplication and subtraction and parity) by predicting what comes next. The problem is broken down into four stages (skill acquisition, skill accumulation, skill composition, and skill as tool use), and the prompts are more detailed that those proposed previously.

There have been ample discussions between the authors and the reviewers, and among the reviewers themselves. Overall, the work is considered relatively narrow although it might inspire further research. The reviewers are not convinced about how the results (which are quite nice) on a single task (difficult perhaps, but limited in complexity) can be generalized and used by the community to "unlock further capabilities of algorithmic prompting" in larger scale contexts. In addition, the technical contributions mainly consist of a set of manually-designed templates. It is unclear what “principle” a general reader can get from the paper.

While one reviewer votes for accept, the other two reviewers consider the work marginally below the acceptance threshold. During discussions, the positive reviewer wrote: “Thank you for your comments. It seems the paper is heading to rejection. I see some merit on how narrow is the class of problems, especially in comparison with the literature. (To be honest, this is all because we are using learning for the tasks. The tasks per see are so small).”


**Summary Of Ac-Reviewer Meeting:**

No one responded to a call for reviewer meeting.  Although the scores (5, 5, 8) suggest disagreements among the reviewers, their comments  indicate otherwise.  During discussions, the positive reviewer wrote: “Thank you for your comments. It seems the paper is heading to rejection. I see some merit on how narrow is the class of problems, especially in comparison with the literature. (To be honest, this is all because we are using learning for the tasks. The tasks per see are so small).”